# COLLAPSED LANGUAGE MODELS PROMOTE FAIRNESS

**Jingxuan Xu**[1*], **Wuyang Chen**[2*], **Linyi Li**[2], **Yao Zhao**[1], **Yunchao Wei**[1†]
[1]Beijing Jiaotong University   [2]Simon Fraser University

## ABSTRACT

To mitigate societal biases implicitly encoded in recent successful pretrained language models, a diverse array of approaches have been proposed to encourage model fairness, focusing on prompting, data augmentation, regularized fine-tuning, and more. Despite the development, it is nontrivial to reach a **principled** understanding of fairness and an effective algorithm that can consistently debias language models. In this work, by rigorous evaluations of *Neural Collapse* – a learning phenomenon happen in last-layer representations and classifiers in deep networks – on fairness-related words, we find that *debiased language models* exhibit *collapsed alignment* between token representations and word embeddings. More importantly, this observation inspires us to design a *principled fine-tuning method* that can effectively improve fairness in a wide range of debiasing methods, while still preserving the performance of language models on standard natural language understanding tasks. We attach our code at https://github.com/Xujxyang/Fairness-NC-main.

## 1 INTRODUCTION

The rise of pre-trained language models (PLMs) has revolutionized natural language processing, greatly enhancing tasks like reasoning and prediction by harnessing the semantic richness of language data. Despite their effectiveness, these models, trained on extensive corpora, often reflect and even intensify societal biases in their training datasets. Such biases manifest in the association of demographic groups with specific roles or capabilities, affecting fairness in applications ranging from legal analytics to hiring processes (Peters et al., 2018; Devlin, 2018; Liu, 2019; Blodgett et al., 2021; Rabelo et al., 2022; Bolukbasi et al., 2016; Caliskan et al., 2017). Thus, it is crucial to address and mitigate these biases to prevent discriminatory practices in downstream applications (Zhao et al., 2019; Webster et al., 2020; Nadeem et al., 2020).

To mitigate societal biases in language models, a substantial array of fairness algorithms has been proposed. On the one hand, people target different learning stages: making language models fair via balanced and augmented training data (Bartl et al., 2020), fine-tuning with regularizations or auxiliary objectives (He et al., 2022; Park et al., 2023), or carefully-tuned prompts (Yang et al., 2023). On the other hand, these debiasing approaches can also be categorized by their awareness of downstream tasks. Task-specific methods fine-tune language models with sensitive annotations (Han et al., 2021b;a; Shen et al., 2021; Ravfogel et al., 2022), while task-agnostic approaches directly debias word embeddings or representations during pretraining (Cheng et al., 2021; Kaneko & Bollegala, 2021; Guo et al., 2022; He et al., 2022). Despite this multitude of efforts, it is challenging to find *common ground* among these methods and *shared properties* of debiased language models. We are thus motivated to ask: *Can we understand and improve the fairness of LMs in principle?*

The recent development of deep learning theory provides fruitful frameworks and tools for us to understand deep neural networks (DNNs). Among them, neural collapse (Papyan et al., 2020) is first observed for classification tasks, and is then analyzed to understand the optimization (Han et al., 2021c; Zhou et al., 2022a) and generalization (Hui et al., 2022; Gao et al., 2023) of DNNs. Meanwhile, the training of generative language models,

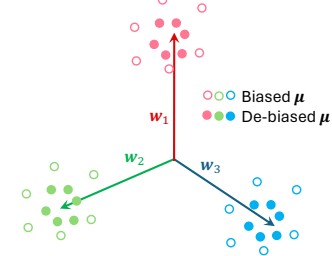

Figure 1: Debiased LMs show more collapsed alignment between classifiers ($w$) and class means ($\mu$). See $(\mathcal{U})\mathcal{NC}_3$ in Table 1.

---

*Equal contribution
†Corresponding author

typically via the **next token prediction task, is essentially a classification problem**. We thus study the fairness of LMs through the lens of neural collapse.

We ask two specific questions:

> ***Q1***: *Do debiased LMs commonly exhibit greater collapse?*
> ***Q2***: *Can we leverage this inductive bias to improve LM fairness in principle?*

Motivated by these two questions, we try to find connections between neural collapse and fairness of language models. We observed, as expected, that fairer language models show more collapsed representations of gender-sensitive words, indicated by greater alignment between classifiers (word embeddings) and class means (token representations) (see Figure 1). This behavior is consistently and implicitly exhibited across a wide range of popular debiasing methods, suggesting a commonly shared perspective on analyzing fairness in language models. This observation further inspires us to explicitly enforce neural collapse to promote fairness in pretrained language models. Since this explicit collapse is principled and does not introduce any customization in fine-tuning or data augmentation, it can be universally applied to enhance many existing fairness approaches, with little implementation or computation overhead. We summarize our contributions below:

- We for the first time comprehensively analyze the relations between neural collapse and fairness in language models.
- Our empirical analysis motivates us to introduce a regularization based on neural collapse, which is extremely simple, and agnostic to any fine-tuning method or augmentation of language data.
- Comprehensive experiments on both intrinsic and extrinsic evaluations demonstrate that our regularization can consistently debias language models. It is orthogonal to a wide range of highly tailored fairness algorithms, and thus can be plug-and-play adopted without sacrificing the models' performance on typical downstream language tasks.

## 2 RELATED WORKS

### 2.1 NEURAL COLLAPSE

A learning phenomenon called *neural collapse* ($\mathcal{NC}$) arises during the terminal phase of training neural networks with cross-entropy (CE) loss for classification tasks (Papyan et al., 2020). It was initially defined by the simultaneous emergence of multiple properties in the model's top-layer features (also referred to as last-layer representations or embeddings) and classifiers, including the variability and geometry of class-wise averaged features, the alignment between features and classifier weights, and the collapse to nearest-neighbor classifier. Since then, $\mathcal{NC}$ has been theoretically analyzed to understand the optimization (Han et al., 2021c; Zhou et al., 2022a) and generalization (Hui et al., 2022; Gao et al., 2023) of DNNs. $\mathcal{NC}$ inspires numerous applications and empirical studies, including transfer learning (Galanti et al., 2021), privacy (Li et al., 2024; 2023b), data imbalance (Fang et al., 2021; Yang et al., 2022), and outlier detection (Liu & Qin, 2023; Wang et al., 2024b;a)

Meanwhile, DNN behaviors related to $\mathcal{NC}$ are also observed in various settings. Several studies have explored $\mathcal{NC}$ under various loss functions. For example, Han et al. (2021c) examined $\mathcal{NC}$ in the context of Mean Squared Error (MSE) loss, while Zhou et al. (2022b) demonstrated that $\mathcal{NC}$ also arises with label smoothing and focal loss. Additionally, researchers like He & Su (2023) and Rangamani et al. (2023) have investigated $\mathcal{NC}$ properties in intermediate layers as well. More recently, Wu & Papyan (2024) observed the linearity of $\mathcal{NC}$ in language models and provided some quantitative explanations for this phenomenon.

### 2.2 FAIRNESS IN LANGUAGE MODELS

Language Models (LMs), extensively studied in academic literature and widely adopted across various applications, have recently raised concerns regarding fairness (Li et al., 2023a; Gallegos et al., 2024). For medium-sized LMs, such as BERT (Devlin, 2018) and Roberta (Liu, 2019), fairness elimination methods have been developed from both data and training perspectives. Given that label imbalance across different demographic groups in the training data is a significant source of bias, a common data processing technique is to balance labels using Counterfactual Data Augmentation (CDA) (Lu et al., 2020; Zmigrod et al., 2019). Some research bridges robustness and fairness by

augmenting a robust training set with techniques such as robust word substitution (Pruksachatkun et al., 2021) and counterfactual logit pairing (Garg et al., 2019). For parameter-efficient methods, GEEP (Fatemi et al., 2021) and Adept (Yang et al., 2023) incorporated gender equality prompts into LLMs using trainable embeddings of occupation names. In addition to retraining, FairBERTa (Qian et al., 2022) showed that fine-tuning language models on the demographic perturbation dataset PANDA can enhance fairness in downstream tasks. For large LMs, such as Llama (Touvron et al., 2023) and GPT (Brown, 2020), recent research not only focuses on fine-tuning the model itself but also emphasizes lightweight post-processing techniques to address fairness concerns, including instruction fine-tuning (Wei et al., 2021; Chung et al., 2024) and prompt engineering (Bubeck et al., 2023; Tamkin et al., 2023).

**Representative Debiased Language Models.** In this paper, we choose to focus on studying the following three representative works, mainly because: 1) They debiased language models from orthogonal perspectives; 2) They all targeted debiasing the BERT model so they can be more fairly compared to each other.

- **Data Preparation: BEC** (Bartl et al., 2020) focused on developing a customized Bias Evaluation Corpus (BEC) via counterfactual data substitution, and studied associations between gender-denoting target words and names of professions (Webster et al., 2018).
- **Fine-tuning Method: Mabel** (He et al., 2022) augmented premises and hypotheses from the natural language inference (NLI) dataset with counterfacts, and applied a contrastive learning objective on gender-balanced entailment pairs to fine-tune BERT.
- **Regularization: ASE** (Park et al., 2023) proposed incorporating the fairness objective into the training process of downstream tasks through two regularization terms (stereotype neutralization and prevention of catastrophic forgetting) beyond the task objective to encourage the fairness.

## 3 DEBIASED LANGUAGE MODELS ARE MORE COLLAPSED

### 3.1 PRELIMINARY: WHY DO LANGUAGE MODELS COLLAPSE?

Suppose the whole vocabulary is the set of word indices $\mathbb{V} = [1, 2, \cdots, C]$. In language models, a sequence of tokens (indices) $\boldsymbol{x}_{1:t} \in \mathbb{V}^t$ are embedded by the word embedding layer $\boldsymbol{E}$ and are forwarded through layers. In the context of next-token prediction for language models, the penultimate token representations $\boldsymbol{h}\left(\boldsymbol{E}(\boldsymbol{x}_{1:t})\right) \in \mathbb{R}^d$ is used to predict the next token $\hat{x}_{t+1} \in \mathbb{V}$ by comparing with classifier weights $\{\boldsymbol{w}_c \in \mathbb{R}^d | c = 1, \cdots, C\}$ and also the bias term:

$$\hat{x}_{t+1} := \underset{c \in \mathbb{V}}{\operatorname{argmax}} \langle \boldsymbol{w}_c, \boldsymbol{h}\left(\boldsymbol{E}\left(\boldsymbol{x}_{1:t}\right)\right)\rangle + \boldsymbol{b}_c \tag{1}$$

We further denote the mean token representation $\boldsymbol{\mu}_c \in \mathbb{R}^d$ whose ground-truth next token is $x_{t+1}^{(s)} = c$. We also focus on the centered means:

$$\boldsymbol{\mu}_c := \frac{1}{N_c} \sum_{s=1}^{S} \sum_{t=1}^{T-1} \boldsymbol{h}\left(\boldsymbol{E}(\boldsymbol{x}_{1:t}^{(s)})\right) \mathbb{I}\left(x_{t+1}^{(s)} = c\right), \quad \overline{\boldsymbol{\mu}} := \frac{1}{C} \sum_{c=1}^{C} \boldsymbol{\mu}_c, \tag{2}$$

where $N_c$ is the number of samples of class $c$, $\mathbb{I}$ is the (binary) indicator function, $S$ is total number of sentences in the dataset, and $T$ is the total number of tokens in a sentence.

We also accumulate the unbiased sample variances:

$$\sigma_c^2 := \frac{1}{N_c - 1} \sum_{s=1}^{S} \sum_{t=1}^{T-1} \left\| \boldsymbol{h}\left(\boldsymbol{E}\left(\boldsymbol{x}_{1:t}^{(s)}\right)\right) - \boldsymbol{\mu}_c \right\|_2^2 \mathbb{I}\left(x_{t+1}^{(s)} = c\right). \tag{3}$$

Neural collapse ($\mathcal{NC}$) (Papyan et al., 2020) is originally observed when neural networks are solving classification problems, where the representations from the penultimate layer and the final classifier weights collapse into certain geometric structures.

**Neural Collapse in Language Models.** From the perspective of token predictions (Eq. 1), we can see that the **pretraining and fine-tuning of language model** are actually **classification tasks**, and the **word embedding layer** $\boldsymbol{E} = \{\boldsymbol{w}_c \in \mathbb{R}^d | c = 1, \cdots, C\}$ is typically the **classifier**. Therefore, we can expect neural collapse also happens in language models, between the mean embedding $\boldsymbol{\mu}_c$ and the word embedding layer $\boldsymbol{E}$.

Specifically, inherited from Papyan et al. (2020), the neural collapse behavior in Language Models can also be defined from four perspectives (Wu & Papyan, 2024):

- $\mathcal{NC}_1$ measures the separability between classes via intra-class variability over inter-class distance: $\mathcal{NC}_1 := \mathbb{E}_{c,c'} \frac{\sigma_c^2 + \sigma_{c'}^2}{2\|\boldsymbol{\mu}_c - \boldsymbol{\mu}_{c'}\|_2^2}$  ($\forall c \neq c'$). A less $\mathcal{NC}_1$ indicates a more collapsed classifier.

- $(\mathcal{G})\mathcal{NC}_2$ measures the separability from the geometric ($\mathcal{G}$) perspective, where the class means tend to become equinorm and equiangular vectors and formulate a simplex known as equiangular tight frame (ETF). This can further be relaxed to account for noises and imbalances in practice: $(\mathcal{G})\mathcal{NC}_2 := \mathbb{E}_{c,c'} \log \left\| \frac{\boldsymbol{\mu}_c - \overline{\boldsymbol{\mu}}}{\|\boldsymbol{\mu}_c - \overline{\boldsymbol{\mu}}\|_2} - \frac{\boldsymbol{\mu}_{c'} - \overline{\boldsymbol{\mu}}}{\|\boldsymbol{\mu}_{c'} - \overline{\boldsymbol{\mu}}\|_2} \right\|^{-1}$  ($\forall c \neq c'$). A less $(\mathcal{G})\mathcal{NC}_2$ indicates more expanded and collapsed class means.

- $(\mathcal{U})\mathcal{NC}_3$ quantifies the alignment between classifiers and class means, termed as "uniform ($\mathcal{U}$) duality". We first measure $\left\langle \frac{\boldsymbol{w}_c}{\|\boldsymbol{w}_c\|_2}, \frac{\boldsymbol{\mu}_c - \overline{\boldsymbol{\mu}}}{\|\boldsymbol{\mu}_c - \overline{\boldsymbol{\mu}}\|_2} \right\rangle$ for each class, and then calculate their standard deviations std($\cdot$) across classes. A less $(\mathcal{U})\mathcal{NC}_3$ indicates class means are more consistently correlated with the classifier, i.e., more collapsed.

- $\mathcal{NC}_4$ simplified the linear projection in Eq. 1 into nearest-class center (NCC) classifier $\underset{c \in \mathbb{V}}{\arg\max} \langle \boldsymbol{w}_c, \boldsymbol{h} \rangle + b_c \rightarrow \underset{c \in \mathbb{V}}{\arg\min} \|\boldsymbol{h} - \boldsymbol{\mu}_c\|_2$ and further measure the token prediction accuracy by nearest neighbors. A greater $\mathcal{NC}_4$ indicates a stronger (more collapsed) NCC classifier.

## 3.2 DEBIASED LMS ARE MORE COLLAPSED THAN BIASED LMS

In the context of fairness of language models, the main idea is to pursue **debiased word embeddings**. For example, people measure the geometry of gender-related tokens in the embedding space (Caliskan et al., 2017; May et al., 2019; Guo & Caliskan, 2021) and try to remove gender-related information of stereotypical words and mitigate biases in the word embedding space. To counteract artifacts from training that leads to the encoding of stereotypes, people even adopt static embeddings (Mikolov et al., 2013; Bolukbasi et al., 2016; Caliskan et al., 2017; Manzini et al., 2019).

Inspired by these previous works, we are expecting to see *more collapse* between the word embedding layer and token representations in *debiased* language models.

### 3.2.1 $\mathcal{NC}$ METRICS IN DEBIASED LMS

We first evaluate $\mathcal{NC}$ metrics in popular debiased language models and compare them with their biased baselines.

**Settings.**  Following previous works, we measure $\mathcal{NC}$ metrics based on different training datasets used in each work, concerning the subset of the whole vocabulary of only gender-related words, dubbed $\mathbb{V}_{\text{gender}}$ (see Appendix A for the full list of words). The detailed datasets used by these works are listed as follows: **Mabel** on SNLI (Bowman et al., 2015) and MNLI (Williams et al., 2017); **ASE** on OntoNotes (Hovy et al., 2006); **BEC** on TinyStories (Eldan & Li, 2023); Among these, only BEC's training dataset (Webster et al., 2018) is relatively small, while the datasets in other works exceed 100K sentences. To ensure meaningful comparisons, we evaluated BEC on a larger dataset TinyStories (Eldan & Li, 2023).

We show this result in Table 1. All methods start from the same pretrained BERT model. However, each work studied the BERT model on its own training data, leading to different $\mathcal{NC}$ measurements for the same BERT model. From Table 1, we can see that debiased language models exhibit neural collapse in certain perspectives: $\mathcal{NC}3$ is consistently improved (minimized) in debiased models, whereas $\mathcal{NC}1/2/4$ are diverging. This indicates that the alignments between token representations ("class means") and debiased word embeddings ("classifier weights") are more consistent, as illustrated in Figure 1. For evaluations of additional models, please refer to Appendix C.

Our explanations are as follows. The neural collapse behavior manifests under certain conditions (Papyan et al., 2020), including: 1) models are trained towards zero training loss; 2) clean labels with balanced classes; 3) the number of classes is not greater than the model's hidden dimension. However, **these conditions are commonly violated in practice**: 1) The training loss is difficult to be minimized to zero due to the complexity of language data; 2) The occurrence of tokens in $\mathbb{V}$ is

Table 1: $(\mathcal{U})\mathcal{NC}_3$ is consistently improved (minimized) on all debiasing methods. Metrics are measured on the subset of the whole vocabulary of only gender-related words $\mathbb{V}_{\text{gender}}$ (Appendix A). In these three groups, each pair of "(BERT, debiased model)" is tested on a customized dataset as detailed in Section 3.2.1, leading to different measurements for the same pretrained BERT model.

| Model | $\mathcal{NC}_1 \downarrow$ | $(\mathcal{G})\mathcal{NC}_2 \downarrow$ | $(\mathcal{U})\mathcal{NC}_3 \downarrow$ | $\mathcal{NC}_4 \uparrow$ | $\mathcal{NC}_1^{(\boldsymbol{w})} \downarrow$ | $(\mathcal{G})\mathcal{NC}_2^{(\boldsymbol{w})} \downarrow$ |
|---|---|---|---|---|---|---|
| BERT | 0.967 | 0.148 | 0.096 | **2.799** | 827.8 | 0.337 |
| MABEL | **0.786** $_{0.181\uparrow}$ | **0.145** $_{0.003\uparrow}$ | **0.070** $_{0.026\uparrow}$ | 2.235 $_{0.564\downarrow}$ | **724.3** $_{103.5\uparrow}$ | **0.331** $_{0.006\uparrow}$ |
| BERT | **2.430** | **0.051** | 0.063 | **1.113** | 1134.7 | **0.364** |
| ASE | 3.008 $_{0.578\downarrow}$ | 0.500 $_{0.449\downarrow}$ | **0.056** $_{0.007\uparrow}$ | 0.023 $_{1.090\downarrow}$ | **382.6** $_{752.1\uparrow}$ | 0.372 $_{0.008\downarrow}$ |
| BERT | **2.358** | **0.152** | 0.062 | **10.44** | 1032.7 | 0.359 |
| BEC | 2.509 $_{0.151\downarrow}$ | 0.185 $_{0.033\downarrow}$ | **0.056** $_{0.006\uparrow}$ | 7.442 $_{2.998\downarrow}$ | **1015.6** $_{17.1\uparrow}$ | **0.335** $_{0.024\uparrow}$ |

highly unbalance due to the nature of languages; 3) Not all language models have greater hidden sizes than the vocabulary size[1]. Therefore, token representations and word embeddings are typically not balanced and well-trained in practical language models, and thus not all perspectives of neural collapse ($\mathcal{NC}1/2/3/4$) can be consistently observed in debiased models. Instead, in the next section, we study how to calibrate these metrics to be more consistent with fairness encoded in models.

### 3.2.2 CALIBRATIONS OF $\mathcal{NC}1$ AND $\mathcal{NC}2$

When we further analyze Table 1 to understand why only $\mathcal{NC}3$ is reduced in debiased LMs as we expected, we find that only $\mathcal{NC}3$ involves the classifier weights $\boldsymbol{w}_c$, but $\mathcal{NC}1/2/4$ all consider class means $\boldsymbol{\mu}_c$. We hypothesize that noises in language data and the complexity of different fine-tuning methods make measurements of $\mathcal{NC}1/2/4$ unstable and inconsistent.

To calibrate these representation-based collapse metrics, we further study replacing class means with classifier weights in $\mathcal{NC}1/2$.

- $\mathcal{NC}_1^{(\boldsymbol{w})} := \mathbb{E}_{c,c'} \frac{\sigma_c^2 + \sigma_{c'}^2}{2\|\boldsymbol{w}_c - \boldsymbol{w}_{c'}\|_2^2} \quad (\forall c \neq c')$.

- $(\mathcal{G})\mathcal{NC}_2^{(\boldsymbol{w})} := \mathbb{E}_{c,c'} \log \|\boldsymbol{w}_c - \boldsymbol{w}_{c'}\|^{-1} \quad (\forall c \neq c')$.

Note that we cannot calibrate $\mathcal{NC}4$ in this way since that will trivially reduce $\mathcal{NC}4$ back to the standard token prediction (Eq. 1).

We show the results of these calibrated $\mathcal{NC}1/2$ in Table 1 (right two columns). Debiased language models exhibit either comparable or more collapsed measurements on $\mathcal{NC}_1^{(\boldsymbol{w})}$ and $(\mathcal{G})\mathcal{NC}_2^{(\boldsymbol{w})}$, showing greater consistency than the uncalibrated $\mathcal{NC}_1$ and $(\mathcal{G})\mathcal{NC}_2$ metrics. This suggests that these calibrated metrics more reliably capture neural collapse in debiased language models.

### 3.3 DEBIASED LMS ARE MORE COLLAPSED IN FAIRNESS-SENSITIVE WORDS

Beyond using different versions of metrics to quantify neural collapse in language models, we further study the impact of different choices of fairness-sensitive words on neural collapse. Our core questions are:

1. Do debiased language models collapse more across the whole vocabulary, or only on fairness-sensitive words?

2. Will the size of subsets of words affect the comparison of $\mathcal{NC}$ metrics?

To answer the above two questions, we calculate $\mathcal{NC}$ metrics on both the whole vocabulary set, and also a random vocabulary subset of the same size as that we used Table 1 (Appendix A). Our gender word list includes 210 female-related words and 215 male-related words.

From Table 2 and Table 3, it is evident that gaps of $\mathcal{NC}$ metrics between BERT and debiased BERT models are much smaller than gaps in Table 1. This indicates that debiased BERT models exhibit more different token representations and word embeddings only on gender-related vocabulary, and they perform much more similarly with BERT on the whole vocabulary. This also explains why

---

[1]Eg. The hidden size of BERT is 768, which is smaller than the vocabulary size of 30,522.

Table 2: $\mathcal{NC}$ metrics of different debiased language models on the whole vocabulary $\mathbb{V}$. Gaps of $\mathcal{NC}$ metrics between BERT and debiased BERT models are much smaller than gaps in Table 1.

| Model | $\mathcal{NC}_1\downarrow$ | $(\mathcal{G})\mathcal{NC}_2\downarrow$ | $(\mathcal{U})\mathcal{NC}_3\downarrow$ | $\mathcal{NC}_4\uparrow$ | $\mathcal{NC}_1^{(\boldsymbol{w})}\downarrow$ | $(\mathcal{G})\mathcal{NC}_2^{(\boldsymbol{w})}\downarrow$ |
|---|---|---|---|---|---|---|
| BERT | 0.340 | 0.248 | **0.056** | 0.712 | 446.8 | 0.407 |
| MABEL | **0.293** $_{0.047\uparrow}$ | **0.240** $_{0.008\uparrow}$ | 0.064 $_{0.006\downarrow}$ | **0.865** $_{0.153\uparrow}$ | **383.7** $_{63.1\uparrow}$ | **0.397** $_{0.010\uparrow}$ |
| BERT | **0.922** | **0.212** | 0.062 | **0.125** | 806.9 | **0.413** |
| ASE | 1.321 $_{0.329\downarrow}$ | 0.417 $_{0.205\downarrow}$ | **0.049** $_{0.013\uparrow}$ | 0.002 $_{0.123\downarrow}$ | **193.1** $_{613.8\uparrow}$ | 0.418 $_{0.005\downarrow}$ |
| BERT | **1.308** | **0.009** | 0.050 | **0.734** | 815.9 | 0.415 |
| BEC | 1.334 $_{0.026\downarrow}$ | 0.011 $_{0.002\downarrow}$ | **0.047** $_{0.003\uparrow}$ | 0.516 $_{0.218\downarrow}$ | **798.4** $_{17.5\uparrow}$ | **0.408** $_{0.007\uparrow}$ |

Table 3: $\mathcal{NC}$ metrics of different debiased language models on the same number of words as used in Table 1 ($\mathbb{V}_{\text{gender}}$) but words are randomly selected. Gaps of $\mathcal{NC}$ metrics between BERT and debiased BERT models are much smaller than gaps in Table 1.

| Model | $\mathcal{NC}_1\downarrow$ | $(\mathcal{G})\mathcal{NC}_2\downarrow$ | $(\mathcal{U})\mathcal{NC}_3\downarrow$ | $\mathcal{NC}_4\uparrow$ | $\mathcal{NC}_1^{(\boldsymbol{w})}\downarrow$ | $(\mathcal{G})\mathcal{NC}_2^{(\boldsymbol{w})}\downarrow$ |
|---|---|---|---|---|---|---|
| BERT | 0.320 | 0.248 | **0.057** | **0.407** | 434.3 | 0.400 |
| MABEL | **0.314** $_{0.006\uparrow}$ | **0.237** $_{0.011\uparrow}$ | 0.060 $_{0.003\downarrow}$ | 1.910 $_{1.503\uparrow}$ | **406.4** $_{27.9\uparrow}$ | **0.392** $_{0.008\uparrow}$ |
| BERT | **1.086** | **0.197** | 0.059 | **0.235** | 859.7 | **0.406** |
| ASE | 1.158 $_{0.072\downarrow}$ | 0.404 $_{0.207\downarrow}$ | **0.046** $_{0.013\uparrow}$ | 0.000 $_{0.235\downarrow}$ | **167.1** $_{692.6\uparrow}$ | 0.436 $_{0.030\downarrow}$ |
| BERT | 1.555 | **0.021** | 0.047 | 0.145 | 866.7 | 0.411 |
| BEC | **1.501** $_{0.054\uparrow}$ | 0.031 $_{0.010\downarrow}$ | **0.044** $_{0.003\uparrow}$ | **1.079** $_{0.934\uparrow}$ | **822.5** $_{44.2\uparrow}$ | **0.405** $_{0.006\uparrow}$ |

neural collapse cannot determine if a language model is debiased or not across the whole vocabulary, as none of the original $\mathcal{NC}$1/2/3/4 metrics show consistent improvement in Table 2. Moreover, as shown in Table 3, this observation also holds on a random subset of vocabulary with matched size with gender-related words in Table 1. Meanwhile, in this context, $\mathcal{NC}$ measurements are generally smaller than those measured on only gender words, which implies that it is generally challenging to learn more separable representations of gender words compared with others insensitive to fairness.

## 4 BIAS MITIGATION VIA ENFORCING EXPLICIT COLLAPSE IN LMS

Motivated by our observations in Section 3, we further ask: can we enforce explicit neural collapse in language models and thus improve their fairness?

We propose minimizing the regularization of language models using $(\mathcal{U})\mathcal{NC}_3$ as an auxiliary objective during fine-tuning:

$$\mathcal{L}_{\mathcal{NC}_3} = \text{std}\left(\left\langle \frac{\boldsymbol{w}_c}{\|\boldsymbol{w}_c\|_2}, \frac{\boldsymbol{\mu}_c - \overline{\boldsymbol{\mu}}}{\|\boldsymbol{\mu}_c - \overline{\boldsymbol{\mu}}\|_2}\right\rangle\right), \quad c \in \mathbb{V}_{\text{gender}}. \tag{4}$$

Although sounds straightforward, this principled approach could potentially be very important in improving the fairness of language models from two perspectives:

1. Our method is simple, principled, and is agnostic to any pretraining or fine-tuning methods for fairness. As we will show in our results, it can be adopted in a wide range of fairness algorithms in a plug-and-play fashion.

2. Our method can avoid manual filtration or augmentation of the underlying language training data.

In the following experiments, we demonstrate that our regularization consistently de-biases language models across different fairness metrics (Sec. 4.1), while still preserving models' language modeling performance (Sec. 4.2). We also provide ablation studies on the strength of this regularization in Appendix B.

### 4.1 ENFORCING $\mathcal{NC}$ PROMOTES FAIRNESS, BOTH INTRINSICALLY AND EXTRINSICALLY

#### 4.1.1 IMPLEMENTATION DETAILS

We conduct experiments following the original settings of each work. **Mabel+$(\mathcal{U})\mathcal{NC}_3$:** We implement Mabel with a batch size of 24, a learning rate of $5 \times 10^{-5}$, and use the Adam optimizer, training it for two epochs. **ASE+$(\mathcal{U})\mathcal{NC}_3$:** ASE is trained for 50 epochs with the Adam optimizer.

The learning rate is set to $2 \times 10^{-5}$, a dropout probability of 0.1 is used, and a batch size of 6 is chosen. **BEC+$(\mathcal{U})\mathcal{NC}_3$**: BEC is trained for three epochs using the Adam optimizer, with a learning rate of $2 \times 10^{-5}$ and a batch size of 16. Due to the short fine-tuning duration, we directly accumulate $\boldsymbol{\mu}_c$ for regularization when computing $(\mathcal{U})\mathcal{NC}_3$.

The evaluation of the fairness of language models can be categorized into addressing intrinsic and extrinsic biases (Li et al., 2023a), which we detail in the following subsections. We also select Sent-Debias (Liang et al., 2020), Context-Debias (Kaneko & Bollegala, 2021), and FairFil (Cheng et al., 2021) as our primary baselines, all of which introduce general-purpose methods for generating debiased representations.

### 4.1.2 INTRINSIC METRICS

The goal of intrinsic debiasing is to reduce bias within model representations before they are applied to downstream tasks, making it task-agnostic. Following previous works (He et al., 2022; Bartl et al., 2020), we consider two popular datasets for intrinsic metrics.

**StereoSet (Nadeem et al., 2020)** evaluates the language model by testing for stereotypical associations. Following He et al. (2022), we concentrate on examples within sentences that pertain to the gender domain. When encountering a partial context sentence, this task presents a fill-in-the-blank challenge where the model must choose from an unrelated word, an anti-stereotypical word, or a stereotypical word. The Language Modeling Score (LM) quantifies the percentage of times the model correctly identifies a relevant word, whether it is stereotypical or anti-stereotypical, or an irrelevant one. Meanwhile, the Stereotype Score (SS) is a fairness-sensitive metric, revealing how often the model shows a preference for the stereotype compared to the anti-stereotype. Lastly, the Idealized Context Association Test (ICAT) score integrates both LM and SS into a single metric, quantifying the balance between language modeling and fairness.

Table 4: Results on StereoSet. $\star$: results are reported in He et al. (2022); We follow previous works to evaluate on their datasets. $\diamond$: the closer to 50, the better. LM: language modeling score, SS: Steoreotype score, ICAT: combined score, defined as $\text{LM} \cdot (\min(\text{SS}, 100 - \text{SS})/50$.

| Model | StereoSet | | |
|---|---|---|---|
| | LM $\uparrow$ | SS $\diamond$ | ICAT $\uparrow$ |
| BERT$^\star$ | 84.17 | 60.28 | 66.86 |
| BERT+DROPOUT$^\star$ | 83.04 | 60.66 | 65.34 |
| BERT+CDA$^\star$ | 83.08 | 59.61 | 67.11 |
| SENT-DEBIAS$^\star$ | 84.20 | 59.37 | 68.42 |
| CONTEXT-DEBIAS$^\star$ | 85.42 | 59.35 | 69.45 |
| FAIRFIL$^\star$ | 44.85 | 50.93 | 44.01 |
| ADEPT | 86.37 | 58.70 | 71.34 |
| MABEL$^\star$ | 84.80 | 56.92 | 73.07 |
| MABEL+$(\mathcal{U})\mathcal{NC}_3$ | 83.55 | **55.38** | **74.55** |
| ASE | 83.83 | 57.33 | 71.54 |
| ASE+$(\mathcal{U})\mathcal{NC}_3$ | 84.06 | **56.36** | **73.37** |
| BEC | 86.02 | 58.30 | 71.73 |
| BEC+$(\mathcal{U})\mathcal{NC}_3$ | 85.95 | **57.89** | **72.38** |

both LM and SS into a single metric, quantifying the balance between language modeling and fairness.

As shown in Table 4, our method consistently improves both SS and ICAT on all fairness methods. With our assistance, Mabel's SS score improved by 1.54 and ICAT increased by 1.48, reaching a value of 74.55. Similarly, ASE achieved a significant ICAT improvement of 1.83.

**BEC-Pro (Bartl et al., 2020)** create a template-based corpus in two languages, English and German, to measure bias in BERT. The sentence templates include a gender-denoting noun phrase, or <person word>, along with a <profession>. Table 5 presents the average association scores between person-related terms (targets) and professions (attributes) before and after applying $(\mathcal{U})\mathcal{NC}_3$ to the debiased model on the GAP corpus (Webster et al., 2018), with Counterfactual Data Substitution (CDS) (Maudslay et al., 2019) applied, highlighting the difference between female and male associations.

We observe a significant reduction in the mean association score difference between female and

Table 5: Results on BEC-Pro. We show the mean association scores between gender words and professions. "Diff" represents the score difference between female and male (smaller the better).

| Model | Female | Male | Diff $\downarrow$ |
|---|---|---|---|
| BERT | -0.0931 | -0.3388 | 0.2457 |
| ADEPT | -0.0056 | 0.0476 | 0.0532 |
| MABLE | -0.0641 | -0.0237 | 0.0404 |
| MABLE+$(\mathcal{U})\mathcal{NC}_3$ | 0.0039 | -0.0038 | **0.0077** |
| ASE | -0.7534 | -0.5125 | 0.2409 |
| ASE+$(\mathcal{U})\mathcal{NC}_3$ | -1.0093 | -0.9613 | **0.0480** |
| BEC | 0.0841 | 0.1349 | 0.0508 |
| BEC+$(\mathcal{U})\mathcal{NC}_3$ | 0.1145 | 0.1435 | **0.0290** |

male after applying our method. This indicates that the model perceives less distinction between these two concepts, implying a reduction in bias. Specifically, Mabel and ASE show a significant order-of-magnitude decrease in the score difference with our method, and BEC also shows over 40% reduction.

### 4.1.3 EXTRINSIC METRICS

Extrinsic metrics (Huang et al., 2019; Smith et al., 2022; Lai et al., 2017) focus on enhancing fairness in downstream tasks, such as sentiment analysis (Mohammad et al., 2018) and machine translation (Levy et al., 2021), by ensuring that models produce consistent outputs across different demographic groups.

**WinoBias Zhao et al. (2018)** is an intra-sentence coreference resolution task designed to assess a system's ability to correctly associate a gendered pronoun with an occupation in both pro-stereotypical and anti-stereotypical contexts. Coreference can be inferred using syntactic cues in Type 1 sentences, or more challenging semantic cues in Type 2 sentences. We fine-tune models on the OntoNotes 5.0 dataset (Hovy et al., 2006) and then evaluate on the WinoBias benchmark. We report the average F1-scores for pro-stereotypical and anti-stereotypical examples, along with two fairness metrics: TPR-1 (Type 1: pro-stereotypical minus anti-stereotypical) and TPR-2 (Type 2: pro-stereotypical minus anti-stereotypical), measured by average F1-scores.

As shown in Table 6, after adding $(\mathcal{U})\mathcal{NC}_3$ as the regularization, all the models exhibit significant improvements on both fairness metrics. Especially, Mabel equipped with our $(\mathcal{U})\mathcal{NC}_3$ improves 6.8 on TPR-1 and more than a 40% improvement on TPR-2 over the Mabel baseline, achieving the best results for our tested models.

Table 6: Average F1-scores on WinoBias, and TPR scores across Winobias categories. 1 = Type 1; 2 = Type 2. A = anti-stereotypical; P = pro-stereotypical. TPR-1 = 1P - 1A; TPR-2 = 2P - 2A.

| Model | 1A ↑ | 1P ↑ | 2A ↑ | 2P ↑ | **TPR-1** ↓ | **TPR-2** ↓ |
|---|---|---|---|---|---|---|
| BERT | 53.96 | 86.57 | 82.20 | 94.67 | 32.79 | 12.48 |
| SENT-DEBIAS | 54.11 | 85.09 | 83.29 | 94.73 | 30.98 | 11.44 |
| CONTEXT-DEBIAS | 59.40 | 85.54 | 83.63 | 93.20 | 26.14 | 9.57 |
| FAIRFIL | 53.24 | 85.77 | 77.37 | 91.40 | 32.43 | 14.03 |
| ADEPT | 62.50 | 84.04 | 87.66 | 91.51 | 21.54 | 3.85 |
| MABEL | 61.21 | 84.93 | 92.78 | 96.20 | 23.73 | 3.41 |
| MABEL+$(\mathcal{U})\mathcal{NC}_3$ | 64.15 | 81.08 | 93.55 | 95.51 | **16.93** | **1.97** |
| ASE | 56.00 | 87.02 | 76.44 | 91.06 | 31.02 | 14.62 |
| ASE+$(\mathcal{U})\mathcal{NC}_3$ | 58.57 | 85.71 | 84.38 | 92.54 | **27.14** | **8.16** |
| BEC | 60.32 | 84.04 | 86.86 | 93.98 | 23.72 | 7.12 |
| BEC+$(\mathcal{U})\mathcal{NC}_3$ | 62.98 | 84.88 | 87.50 | 94.40 | **21.91** | **6.90** |

**Bias-in-Bios (De-Arteaga et al., 2019)** is a third-person biography dataset labeled by both occupation and gender. We fine-tune the encoder with the linear classification layer to predict an individual's occupation from their biography. During evaluation, we present the overall accuracy for the task, as well as the accuracy segmented by gender. Additionally, we use two widely adopted fairness metrics De-Arteaga et al. (2019); Ravfogel et al. (2020): 1) $GAP_{TPR}$, which captures the disparity in true positive rates (TPR) between male- and female-labeled instances; and 2) $GAP_{RMS}$, which represents the root-mean-square (RMS) of TPR gaps across different occupation categories ($C$), the closer their score is to 0, the better. Their formula is as follows:

$$GAP_{TPR} = |TPR_M - TPR_F|, \qquad GAP_{RMS} = \sqrt{\frac{1}{|C|} \sum_{y \in C} (GAP_{TPR,y})^2}. \qquad (5)$$

In Table 7, we observe that introducing $(\mathcal{U})\mathcal{NC}_3$ during the fine-tuning process improves the model's performance on these two TPR metrics. Additionally, it enhances the model's accuracy in predicting female-related occupations, helping BEC achieve an accuracy of 85.17 for female predictions.

Furthermore, we also conducted validation on the Bias-NLI (Dev et al., 2020), and the results can be found in the Appendix D.

Table 7: Fine-tuning results on Bias-in-Bios.

| Model | Acc. (All) ↑ | Acc. (M) ↑ | Acc. (F) ↑ | GAP TPR ↓ | GAP RMS ↓ |
|---|---|---|---|---|---|
| BERT | 84.14 | 84.69 | 83.50 | 1.189 | 0.144 |
| SENT-DEBIAS | 83.56 | 84.10 | 82.92 | 1.180 | 0.144 |
| CONTEXT-DEBIAS | 83.67 | 84.08 | 83.18 | 0.931 | 0.137 |
| FAIRFIL | 83.18 | 83.52 | 82.78 | 0.746 | 0.142 |
| ADEPT | 84.07 | 83.64 | 84.58 | 0.945 | 0.121 |
| MABLE | 84.85 | 84.92 | 84.34 | 0.599 | 0.132 |
| MABLE+$(\mathcal{U})\mathcal{NC}_3$ | 84.30 | 84.09 | 84.54 | **0.455** | **0.126** |
| ASE | 84.63 | 84.19 | 85.14 | 0.949 | 0.127 |
| ASE+$(\mathcal{U})\mathcal{NC}_3$ | 84.55 | 84.21 | 84.95 | **0.740** | **0.126** |
| BEC | 84.43 | 83.99 | 84.95 | 0.954 | **0.124** |
| BEC+$(\mathcal{U})\mathcal{NC}_3$ | 84.66 | 84.23 | 85.17 | **0.938** | 0.128 |

## 4.2 ENFORCING $\mathcal{NC}$ PRESERVES LM PERFORMANCE ON NLU TASKS

We also evaluate language models fine-tuned with our method on general natural language understanding (NLU) tasks. As shown in Table 8, fine-tuning with Eq. 4 preserves the performance of language models on general tasks with minimal difference from each baseline model. This indicates that our method can debias language models without catastrophically forgetting and sacrificing their pretrained knowledge.

## 5 ABLATION STUDIES AND VISUALIZATIONS

**Fairness Can Be Improved Solely via $\mathcal{NC}$.** As we have seen that debiased language models can implicitly become more collapsed, here we further provide an ablation study to verify that the effectiveness of our regularization is not because of any external debiasing algorithm, namely, by *only* adopting our NC-based regularization (Eq. 4) we can already encourage fairness in language models. We perform this ablation study in the standard language model fine-tuning for masked language modeling (MLM), where pronouns in the input sentence are replaced with [MASK] tokens. The model is then trained to predict the correct gendered pronoun for each [MASK] token in the masked sentence. The training objective is to minimize the cross-entropy loss between the original pronouns and the predicted logits corresponding to the [MASK] tokens. The MLM loss is denoted as $\mathcal{L}_{\text{MLM}}$:

$$\mathcal{L}_{\text{MLM}} = \frac{1}{|M|} \sum_{m \in \text{masked}}^{M} CE(\boldsymbol{W}^\top \boldsymbol{h}_m, x_m) \tag{6}$$

where $CE$ denotes the cross entropy loss, and $\boldsymbol{h}_m \in \mathbb{R}^d$ is the last hidden state of the masked token $x_m$. $\boldsymbol{W} \in \mathbb{R}^{d,C}$ is a linear layer for the MLM task.

We fine-tune the language model using $\mathcal{L}_{\text{MLM}}$ alone, and compare against the fine-tuning with the addition of $\mathcal{L}_{\mathcal{NC}_3}$. We fine-tune models for three epochs. This allows us to more clearly observe the gains brought by $(\mathcal{U})\mathcal{NC}_3$. We evaluate the model on StereoSet using intrinsic metrics, and as shown in Table 9, it is evident that not only does it help to reduce bias in the model, but also enhances the language capabilities. Notably, we achieve a significant improvement of 4.29 on the ICAT metric.

**Visualizations of Debiased Token Representations.** In addition to comparing quantitative results, we also conducted t-SNE visualizations of models' logits to analyze the two training methods. Specifically, we visualize two opposing word pairs: ("Herself", "Himself") and ("Mother", "Father"). As shown in Figure 2, with the original training method, words of the same gender tend to cluster together (e.g., "Himself" and "Father"). However, after introducing $(\mathcal{U})\mathcal{NC}_3$, we can find that: 1) token representations from different classes (words) are more clearly separated to each other;

Table 8: Fine-tuning results on the GLUE benchmark: $(\mathcal{U})\mathcal{NC}_3$ preserves LM performance while encouraging fairness. BERT-NLI: BERT fine-tuned on NLI data then fine-tuned on GLUE. Bold numbers indicate the best for different metrics.

| Model | CoLA ↑ (mcc.) | SST-2 ↑ (acc.) | MRPC ↑ (f1/acc.) | QQP ↑ (acc./f1) | MNLI ↑ (acc.) | QNLI ↑ (acc.) | RTE ↑ (acc.) | STS-B ↑ (pears./spear.) |
|---|---|---|---|---|---|---|---|---|
| BERT | 56.5 | 92.3 | 89.5/85.3 | 90.7/87.5 | 84.3 | 92.2 | 65.0 | 88.4/88.2 |
| BERT-NLI | 58.6 | 93.6 | 89.4/85.1 | 90.4/86.8 | 83.3 | 89.0 | 69.0 | 88.3/87.9 |
| SENT-DEBIAS | 50.5 | 89.1 | 87.5/81.6 | 87.5/90.7 | 83.9 | 91.4 | 63.2 | 88.1/87.9 |
| CONTEXT-DEBIAS | 55.2 | 92.0 | 85.1/77.5 | 90.7/87.4 | 84.6 | 89.9 | 57.0 | 88.4/88.1 |
| FAIRFIL | 55.5 | 92.4 | 87.5/80.6 | 91.2/88.1 | 84.8 | 91.3 | 63.2 | 88.4/88.1 |
| ADEPT | 57.3 | 92.4 | 88.0/82.4 | - - / - - | 84.1 | 91.0 | 61.4 | 85.8/85.7 |
| MABEL | 57.8 | 92.2 | 89.5/85.0 | 91.2/88.1 | 84.5 | 91.6 | 64.3 | 89.6/89.2 |
| MABEL+$(\mathcal{U})\mathcal{NC}_3$ | 56.3 | 92.1 | 90.5/86.5 | 91.0/88.0 | 84.5 | 91.1 | 60.7 | 89.1/88.7 |
| ASE | 54.9 | 92.9 | 87.3/80.9 | - - / - - | 84.4 | 91.2 | 59.3 | 88.4/88.1 |
| ASE+$(\mathcal{U})\mathcal{NC}_3$ | 55.3 | 92.8 | 87.5/81.4 | - - / - - | 84.6 | 91.4 | 61.7 | 88.2/87.9 |
| BEC | 55.5 | 92.2 | 88.8/83.8 | - - / - - | 84.9 | 91.5 | 65.7 | 89.2/88.9 |
| BEC+$(\mathcal{U})\mathcal{NC}_3$ | 55.7 | 92.5 | 88.7/83.6 | - - / - - | 84.9 | 91.6 | 66.4 | 89.2/88.9 |

Table 9: Fine-tuning BERT with only mask-token predictions (Eq. 6) and $(\mathcal{U})\mathcal{NC}_3$ (Eq. 4). Metrics are consistent with Table 4: "LM" for language modeling score, "SS" for Steoreotype score, "ICAT" for combined score, defined as $\text{LM} \cdot (\min(\text{SS}, 100 - \text{SS}))/50$.

| Model | StereoSet | | |
|---|---|---|---|
| | LM ↑ | SS ⬦ | ICAT ↑ |
| BERT+$\mathcal{L}_{\text{MLM}}$ | 72.85 | 58.65 | 60.24 |
| BERT+$\mathcal{L}_{\text{MLM}}$+$\mathcal{L}_{\mathcal{NC}_3}$ | **75.43** | **57.23** | **64.53** |

2) tokens from each word are more clustered. This indicates that intra-class variances have been significantly reduced, thus gender-related words are more collapsed to their class means.

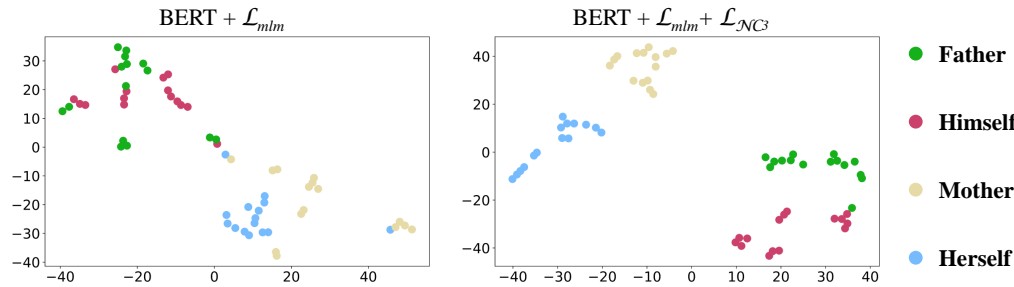

Figure 2: t-SNE plots of logits of two models in Table 9. We collect 15 samples each of "Herself," "Himself," "Mother," and "Father."

## 6 CONCLUSION

Aiming to provide a principled understanding and improvement of debiasing language models, we try to find connections between language models and neural collapse for fairness purposes. We demonstrated that debiased models exhibit more collapsed token representations, especially for fairness-sensitive words, leading to better alignment with word embeddings. Leveraging this insight, we introduced a principled regularization method based on neural collapse that can consistently improve fairness in language models across various tasks without sacrificing performance. Our method is simple, effective, and applicable to a wide range of debiasing techniques, providing a robust tool for enhancing fairness in language models. We expect our understanding and fine-tuning to become a principled method for debiasing language models.

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

## A GENDER WORDS

**Male:** countryman, fraternal, wizards, manservant, fathers, divo, actor, bachelor, papa, dukes, barman, countrymen, brideprice, hosts, airmen, andropause, penis, prince, governors, abbot, men, widower, gentlemen, sorcerers, sir, bridegrooms, baron, househusbands, gods, nephew, widowers, lord, brother, grooms, priest, adultors, andrology, bellboys, his, marquis, princes, emperors, stallion, chairman, monastery, priests, boyhood, fellas, king, dudes, daddies, manservant, semen, spokesman, tailor, cowboys, dude, bachelors, barbershop, emperor, daddy, masculism, guys, enchanter, guy, fatherhood, androgen, cameramen, godfather, strongman, god, patriarch, uncle, chairmen, sir, brotherhood, host, testosterone, husband, dad, steward, males, cialis, spokesmen, pa, beau, stud, bachelor, wizard, sir, nephews, fathered, bull, beaus, councilmen, landlords, grandson, fiances, stepfathers, horsemen, grandfathers, adultor, schoolboy, rooster, grandsons, bachelor, cameraman, dads, him, master, lad, policeman, monk, actors, salesmen, boyfriend, councilman, fella, statesman, paternal, chap, landlord, brethren, lords, blokes, fraternity, bellboy, duke, balletdancer, dudes, fiance, colts, husbands, suitor, paternity, he, businessman, masseurs, hero, deer, busboys, boyfriends, kings, brothers, masters, stepfather, grooms, son, studs, cowboy, mentleman, sons, baritone, salesman, paramour, malehost, monks, menservants, mr., headmasters, lads, congressman, airman, househusband, priest, barmen, barons, abbot, handyman, beard, fraternities, stewards, colt, czar, stepsons, himself, boys, lions, gentleman, penis, his, masseur, bulls, uncles, bloke, beards, hubby, lion, sorcerer, macho, father, gays, male, waiters, sperm, prostate, stepson, prostaticutricle, businessmen, heir, waiter, headmaster, man, governor, god, bridegroom, grandpa, groom, dude, gay, gents, boy, grandfather, gelding, paternity, roosters, prostaticutricle, priests, manservants, stailor, busboy, heros.

**Female:** countrywoman, sororal, witches, maidservant, mothers, diva, actress, spinster, mama, duchesses, barwoman, countrywomen, dowry, hostesses, airwomen, menopause, clitoris, princess, governesses, abbess, women, widow, ladies, sorceresses, madam, brides, baroness, housewives, goddesses, niece, widows, lady, sister, nun, adultresses, obstetrics, bellgirls, marchioness, princesses, empresses, mare, chairwoman, convent, priestesses, girlhood, gals, mommies, maid, female ejaculation, spokeswoman, seamstress, cowgirls, chick, hairsalon, empress, mommy, feminism, enchantress, gal, motherhood, estrogen, camerawomen, godmother, strongwoman, goddess, matriarch, aunt, chairwomen, ma'am, sisterhood, hostess, estradiol, wife, mom, stewardess, females, viagra, spokeswomen, ma, belle, minx, maiden, witch, miss, nieces, mothered, cow, belles, councilwomen, landladies, granddaughter, fiancees, stepmothers, horsewomen, grandmothers, adultress, schoolgirl, hen, granddaughters, bachelorette, camerawoman, moms, mistress, lass, policewoman, saleswomen, girlfriend, councilwoman, stateswoman, maternal, wenches, sorority, ballerina, chicks, fiancee, fillies, suitress, maternity, she, businesswoman, masseuses, heroine, doe, busgirls, girlfriends, queens, sisters, mistresses, stepmother, daughter, minxes, cowgirl, mezzo, saleswoman, nuns, maids, mrs., headmistresses, lasses, congresswoman, airwoman, housewife, priestess, barwomen, baronesses, abbesses, handywoman, toque, sororities, stewardesses, filly, czarina, stepdaughters, herself, girls, lionesses, vagina, hers, masseuse, aunts, wench, toques, heiress, waitress, headmistress, bride, grandma, lesbian, girl, grandmother, hens, uterus, maidservants, seamstress', busgirl, heroines.

## B ABLATION STUDY ON $\mathcal{NC}_3$ REGULARIZATION

Our method is simple and principled. It adds our $\mathcal{NC}_3$ constraint to the original training loss used in each work, as shown in the following formula:

$$\mathcal{L}_{total} := \mathcal{L}_{original} + \alpha \cdot \mathcal{L}_{\mathcal{NC}_3}, \tag{7}$$

where $\mathcal{L}_{original}$ represents the original loss from the respective study, and $\mathcal{L}_{\mathcal{NC}_3}$ denotes our regularization term.

From Table 10, it's clear that as $\alpha$ increases, the SS metric generally improves, indicating that the model becomes more biased. However, a more appropriate value of $\alpha$ can strike a better balance between language quality and bias, resulting in a higher ICAT score.

## C $\mathcal{NC}$ EVALUATIONS FOR MORE LANGUAGE MODELS

In addition to the three debiasing methods tested in Section 3, we supplement our analysis here with experiments from two popular models: BERT base and RoBERTa base. The results from these models

Table 10: StereoSet results on different hyper-parameter settings of $\alpha$. Metrics are consistent with Table 4: "LM" for language modeling score, "SS" for Steoreotype score, "ICAT" for combined score, defined as LM $\cdot (\min(\text{SS}, 100 - \text{SS}))/50$.

| | ASE+$(\mathcal{U})\mathcal{NC}_3$ | | | | Mabel+$(\mathcal{U})\mathcal{NC}_3$ | | | BEC+$(\mathcal{U})\mathcal{NC}_3$ | | |
|---|---|---|---|---|---|---|---|---|---|---|
| | LM $\uparrow$ | SS $\diamond$ | ICAT $\uparrow$ | | LM $\uparrow$ | SS $\diamond$ | ICAT $\uparrow$ | LM $\uparrow$ | SS $\diamond$ | ICAT $\uparrow$ |
| $\alpha = 1$ | 83.78 | 57.38 | 71.41 | $\alpha = 10$ | **83.57** | 55.53 | 74.33 | **86.02** | 58.07 | 72.12 |
| $\alpha = 3$ | **84.06** | 56.36 | **73.37** | $\alpha = 30$ | 83.54 | 55.63 | 73.88 | 85.95 | **57.89** | **72.38** |
| $\alpha = 5$ | 82.96 | **55.84** | 73.27 | $\alpha = 50$ | 83.55 | **55.38** | **74.55** | 85.70 | 58.92 | 71.44 |

(Table 11, 12, and 13) are consistent with the observations made in Section 3. The three additional model setups are as follows: **Adept** on News-Commentary v15 (Tiedemann, 2012); **Roberta** and **FairBERTa** on PANDA (Qian et al., 2022); **Mabel** under Roberta as the backbone.

Table 11: The performance of more debiasing methods on $\mathcal{NC}$ metrics varies across datasets on the gender words ($\mathbb{V}_{\text{gender}}$).

| Model | $\mathcal{NC}_1 \downarrow$ | $(\mathcal{G})\mathcal{NC}_2 \downarrow$ | $(\mathcal{U})\mathcal{NC}_3 \downarrow$ | $\mathcal{NC}_4 \uparrow$ | $\mathcal{NC}_1^{(\boldsymbol{w})} \downarrow$ | $(\mathcal{G})\mathcal{NC}_2^{(\boldsymbol{w})} \downarrow$ |
|---|---|---|---|---|---|---|
| BERT | 2.090 | 0.088 | 1.317 | **0.169** | 1197.6 | **0.359** |
| ADEPT (YANG ET AL., 2023) | **1.941** | **0.072** | **0.074** | 0.118 | **1143.5** | 0.364 |
| ROBERTA | **0.868** | **0.010** | 0.066 | **0.513** | 11.53 | **1.614** |
| FAIRBERTA (QIAN ET AL., 2022) | 1.287 | 0.661 | **0.038** | 0.000 | **5.883** | 1.698 |
| ROBERTA | **0.443** | 0.059 | 0.062 | 0.250 | 10.22 | 1.488 |
| MABEL-ROBERTA (QIAN ET AL., 2022) | 0.513 | **0.013** | **0.058** | 0.250 | **9.878** | **1.487** |

Table 12: $\mathcal{NC}$ metrics of more debiased language models on the whole vocabulary ($\mathbb{V}$).

| Model | $\mathcal{NC}_1 \downarrow$ | $(\mathcal{G})\mathcal{NC}_2 \downarrow$ | $(\mathcal{U})\mathcal{NC}_3 \downarrow$ | $\mathcal{NC}_4 \uparrow$ | $\mathcal{NC}_1^{(\boldsymbol{w})} \downarrow$ | $(\mathcal{G})\mathcal{NC}_2^{(\boldsymbol{w})} \downarrow$ |
|---|---|---|---|---|---|---|
| BERT | 1.235 | 0.165 | **0.064** | 0.013 | 946.2 | **0.413** |
| ADEPT (YANG ET AL., 2023) | **1.133** | **0.149** | 0.066 | 0.012 | **889.2** | 0.420 |
| ROBERTA | **0.405** | 0.116 | 0.065 | **0.647** | 8.334 | **1.617** |
| FAIRBERTA (QIAN ET AL., 2022) | 0.766 | 0.520 | **0.036** | 0.000 | **4.545** | 1.732 |
| ROBERTA | 0.246 | 0.135 | **0.060** | 0.369 | 6.219 | 1.567 |
| MABEL-ROBERTA (QIAN ET AL., 2022) | **0.242** | **0.061** | 0.061 | **0.568** | **5.363** | 1.567 |

Table 13: $\mathcal{NC}$ metrics of more debiased language models on the same number of words as used in Table 11 ($\mathbb{V}_{\text{gender}}$) but words are randomly selected.

| Model | $\mathcal{NC}_1 \downarrow$ | $(\mathcal{G})\mathcal{NC}_2 \downarrow$ | $(\mathcal{U})\mathcal{NC}_3 \downarrow$ | $\mathcal{NC}_4 \uparrow$ | $\mathcal{NC}_1^{(\boldsymbol{w})} \downarrow$ | $(\mathcal{G})\mathcal{NC}_2^{(\boldsymbol{w})} \downarrow$ |
|---|---|---|---|---|---|---|
| BERT | 1.292 | 0.158 | 0.065 | 0.002 | 974.7 | **0.409** |
| ADEPT (YANG ET AL., 2023) | **1.111** | **0.150** | **0.064** | 0.002 | **888.6** | 0.423 |
| ROBERTA | **0.407** | 0.116 | 0.067 | **0.811** | 8.416 | **1.618** |
| FAIRBERTA (QIAN ET AL., 2022) | 0.728 | 0.511 | **0.036** | 0.000 | **4.414** | 1.733 |
| ROBERTA | **0.217** | 0.146 | **0.058** | 0.162 | 5.653 | **1.561** |
| MABEL-ROBERTA (QIAN ET AL., 2022) | 0.237 | **0.063** | 0.064 | **0.280** | **5.267** | 1.572 |

## D    EXTRA EXTRINSIC METRICS

**Bias-NLI Dev et al. (2020)** is an NLI dataset composed of neutral sentence pairs, systematically generated by populating sentence templates with a gendered word and an occupation that has a strong gender association (e.g., "The *woman* ate a bagel" "The *nurse* ate a bagel"). Bias is measured as a deviation from neutrality and is evaluated using three metrics: Net Neutral (NN), Fraction Neutral (FN), and Threshold:$\tau$ (T:$\tau$). Specifically, 1) NN represents the mean probability assigned to the

neutral label across all entailment pairs; 2) FN calculates the proportion of sentence pairs classified as neutral; and 3) Threshold:$\tau$ (T:$\tau$) is a hyperparameter that determines the proportion of entailment pairs for which the probability of being neutral exceeds a given threshold. In this paper, $\tau$ is set to 0.5 and 0.7. And a bias-free model would achieve a score as 1 across all three metrics. NN and FN are defined in the following manner:

$$NN = \frac{1}{M} \sum_{i=1}^{M} n_i, \qquad FN = \frac{1}{M} \sum_{i=1}^{M} n_i 1[n_i = max\{e_i, n_i, c_i\}], \qquad (8)$$

We fine-tune the model on SNLI and assess its performance on Bias-NLI during inference. After incorporating $(\mathcal{U})\mathcal{NC}_3$ during fine-tuning, we find that in such a rare-class classification scenario, the improvement is relatively limited. However, it generally helps the model maintain its performance, leading to some gains in accuracy across various metrics for ASE and BEC. Specific details can be found in Table 14.

Table 14: Results on Bias-NLI. We fine-tune the models on SNLI and then evaluate on Bias-NLI.

| Model | TN ↑ | FN ↑ | T:0.5 ↑ | T:0.7 ↑ |
|---|---|---|---|---|
| BERT | 0.799 | 0.879 | 0.874 | 0.798 |
| SENT-DEBIAS | 0.793 | 0.911 | 0.897 | 0.788 |
| CONTEXT-DEBIAS | 0.858 | 0.906 | 0.902 | 0.857 |
| FAIRFIL | 0.829 | 0.883 | 0.846 | 0.845 |
| ADEPT | 0.841 | 0.937 | 0.934 | 0.866 |
| MABLE | **0.900** | 0.977 | 0.974 | **0.935** |
| MABLE+$(\mathcal{U})\mathcal{NC}_3$ | 0.882 | 0.977 | 0.974 | 0.919 |
| ASE | 0.886 | 0.974 | 0.971 | 0.932 |
| ASE+$(\mathcal{U})\mathcal{NC}_3$ | **0.910** | 0.974 | **0.973** | **0.948** |
| BEC | **0.933** | 0.955 | 0.955 | 0.934 |
| BEC+$(\mathcal{U})\mathcal{NC}_3$ | 0.883 | **0.980** | **0.978** | **0.943** |

# E $\mathcal{NC}$ EVALUATIONS FOR NEUTRAL WORDS

In addition to gender-related words such as "female" and "male," there are also neutral words. To assess the impact of these neutral words, we conducted ablation experiments. The experimental setup is as follows:

- **Gender words + Neutral words:** including 425 gender-related words from Appendix A and 208 neutral words mentioned below.

- **Gender words + Random words:** including 425 gender-related words from Appendix A and 208 Random words.

**Neutral:** aerobics, adventurer, apparel, aggressive, assistant, tycoon, baker, warrior, bathing, ambitious, beautiful, trucker, beauty, welder, blonde, strong, bookkeeper, terrorist, ca, soldier, cashier, astronomer, chatty, sniper, cheerleader, skipper, cheerleading, banker, clerk, baseball, cocktail, sergeant, cooking, bodyguard, counselor, boss, crafting, boxer, cute, cabbie, dancer, captain, educator, cardiologist, emotional, carpenter, flirt, ceo, flirtatious, chairperson, flower, chancellor, gossip, chef, graceful, colonel, hairdresser, commander, hairdryer, conductor, homemaker, police, hooker, custodian, housekeeper, dentist, housekeepers, detective, housework, diplomat, hula, doctor, indoor, driving, jealousy, drummer, jewelry, economist, kawaii, electrician, laundering, engineer, librarian, engineering, librarians, entrepreneur, lotion, lovely, firefighter, marvelous, footballer, mirror, gambler, moisturizer, gamer, nanny, gangster, neat, geek, nurse, geeks, nursery, gentle, nurses, guitarist, nurturing, industrialist, parenting, inventor, passive, investigator, pink, laborer, pretty, lawyer, receptionist, leader, ribbon, lieutenant, romance, lifeguard, romantic, magistrate, secretary, manager, selfie, marshal, server, mathematician, sew, mechanic, sewing, muscle, shopping, muscular, smoothie, owner,

soft, philosopher, softball, physicist, stylist, pilot, submissive, plumber, sweet, politician, tailor, president, tall, professor, teacher, programmer, thin, rugby, violinist, sailor, waiter, science, weak, scientist, yoga, sculptor, hysterical, blue, makeup, football, executive, management, professional, corporation, salary, office, business, career, home, parents, children, family, cousins, marriage, wedding, relatives, math, algebra, geometry, calculus, equations, computation, numbers, addition, poetry, art, dance, literature, novel, symphony, drama, sculpture, science, technology, physics, chemistry, Einstein, NASA, experiment, astronomy, Shakespeare.

As shown in Table 15 and Table 16, results of the two different word combinations on the $\mathcal{NC}$ metrics are very similar, which further supports the rationale for considering only gender-related words in our method. However, upon closer inspection, we can observe that when using neutral words, the metric values tend to be closer to those obtained with only gender words (Table 1 and Table 11), whereas the values with random words are more aligned with those generated by random words (Table 3 and Table 13). This further supports the validity of the $\mathcal{NC}$ metric.

Table 15: $\mathcal{NC}$ metrics of different debiased language models on Gender words + Neutral words.

| Model | $\mathcal{NC}_1 \downarrow$ | $(\mathcal{G})\mathcal{NC}_2 \downarrow$ | $(\mathcal{U})\mathcal{NC}_3 \downarrow$ | $\mathcal{NC}_4 \uparrow$ | $\mathcal{NC}_1^{(\boldsymbol{w})} \downarrow$ | $(\mathcal{G})\mathcal{NC}_2^{(\boldsymbol{w})} \downarrow$ |
|---|---|---|---|---|---|---|
| BERT | 0.894 | **0.139** | **0.066** | **1.634** | 790.3 | 0.337 |
| MABEL (HE ET AL., 2022) | **0.698** | 0.141 | 0.069 | 1.421 | **687.6** | **0.330** |
| BERT | **2.182** | **0.073** | 0.064 | **0.684** | 1155.1 | 0.354 |
| ASE (PARK ET AL., 2023) | 2.975 | 0.566 | **0.054** | 0.013 | **324.3** | 0.363 |
| BERT | **1.988** | **0.145** | 0.060 | **6.260** | 1033.0 | 0.351 |
| BEC (BARTL ET AL., 2020) | 2.105 | 0.183 | **0.055** | 4.469 | **1014.7** | **0.346** |
| BERT | 2.275 | 0.071 | 0.070 | **0.099** | 1267.4 | **0.351** |
| ADEPT (YANG ET AL., 2023) | **2.104** | **0.053** | 0.070 | 0.070 | **1206.4** | 0.357 |
| ROBERTA | **0.737** | **0.018** | 0.065 | **0.693** | 10.48 | **1.621** |
| FAIRBERTA (QIAN ET AL., 2022) | 1.138 | 0.632 | **0.035** | 0.000 | **5.459** | 1.706 |
| ROBERTA | **0.317** | 0.075 | 0.065 | **0.155** | 6.986 | 1.553 |
| MABEL-ROBERTA (QIAN ET AL., 2022) | 0.333 | **0.069** | 0.065 | 0.134 | **5.573** | 1.553 |

Table 16: $\mathcal{NC}$ metrics of different debiased language models on Gender words + Random words.

| Model | $\mathcal{NC}_1 \downarrow$ | $(\mathcal{G})\mathcal{NC}_2 \downarrow$ | $(\mathcal{U})\mathcal{NC}_3 \downarrow$ | $\mathcal{NC}_4 \uparrow$ | $\mathcal{NC}_1^{(\boldsymbol{w})} \downarrow$ | $(\mathcal{G})\mathcal{NC}_2^{(\boldsymbol{w})} \downarrow$ |
|---|---|---|---|---|---|---|
| BERT | 0.466 | 0.225 | **0.061** | **1.137** | 533.5 | 0.390 |
| MABEL (HE ET AL., 2022) | **0.413** | **0.219** | 0.065 | 1.132 | **475.6** | **0.384** |
| BERT | **1.534** | **0.142** | 0.064 | **0.644** | 943.3 | **0.391** |
| ASE (PARK ET AL., 2023) | 2.076 | 0.445 | **0.055** | 0.012 | **288.17** | 0.397 |
| BERT | **1.742** | **0.052** | 0.083 | **5.721** | 891.8 | 0.399 |
| BEC (BARTL ET AL., 2020) | 1.903 | 0.093 | **0.051** | 4.115 | 892.6 | **0.389** |
| BERT | 1.555 | 0.140 | **0.068** | **0.093** | 1043.0 | **0.394** |
| ADEPT (YANG ET AL., 2023) | **1.473** | **0.117** | 0.069 | 0.067 | **1010.9** | 0.397 |
| ROBERTA | **0.515** | **0.093** | 0.068 | **0.256** | 9.151 | **1.631** |
| FAIRBERTA (QIAN ET AL., 2022) | 0.909 | 0.552 | **0.037** | 0.000 | **5.032** | 1.731 |
| ROBERTA | **0.255** | 0.135 | 0.063 | 0.591 | 6.508 | **1.564** |
| MABEL-ROBERTA (QIAN ET AL., 2022) | 0.260 | **0.040** | **0.062** | **1.617** | **5.595** | 1.567 |

## F  DETAILED EXPLANATION OF BEC-PRO

The association score accurately measures the correlation between a <person word> and a <profession>, with smaller values indicating a weaker correlation. However, it is important to note that the association score itself does not directly indicate gender bias. For instance, both male and female associations could be weak, but the difference between these scores can reveal gender bias, where the profession may show varying inclinations toward males and females. When examining the disparity in association scores between <female> and <male> with respect to a particular profession, this difference can be interpreted as a measure of gender bias in occupational associations. As such, this metric serves as a more relevant and intrinsic measure of gender bias, specifically addressing how professions are biased in their associations with gender.

To aid in understanding, we conducted an experiment in Table 17, which $(\mathcal{U})\mathcal{NC}_3$ was optimized in BEC using different sets of words. The results demonstrate that both gender and profession words can reduce the association scores. However, regularizing $(\mathcal{U})\mathcal{NC}_3$ using gender-specific words leads to a reduced distinction between female and male associations (i.e., a decrease in the difference), whereas regularizing with profession words actually escalates the gender bias.

Table 17: More results on BEC-Pro. We train BEC by utilizing different word lists (gender/profession) for $(\mathcal{U})\mathcal{NC}_3$. $\alpha = 10$ (Eq. 7)

| Model | Female | Male | Diff $\downarrow$ |
|---|---|---|---|
| BEC | 0.0841 | 0.1349 | 0.0508 |
| BEC+$(\mathcal{U})\mathcal{NC}_3$ ON GENDER | 0.0632 $\downarrow$ | 0.1103 $\downarrow$ | 0.0471 $\downarrow$ |
| BEC+$(\mathcal{U})\mathcal{NC}_3$ ON PROFESSION | 0.0429 $\downarrow$ | 0.1200 $\downarrow$ | 0.0771 $\uparrow$ |

## G    ABSOLUTE CHANGES OF $\mathcal{NC}$ AFTER DEBIASING

From Figure 3, it is evident that $\mathcal{NC}$ is more sensitive to fairness-related words, such as gender.

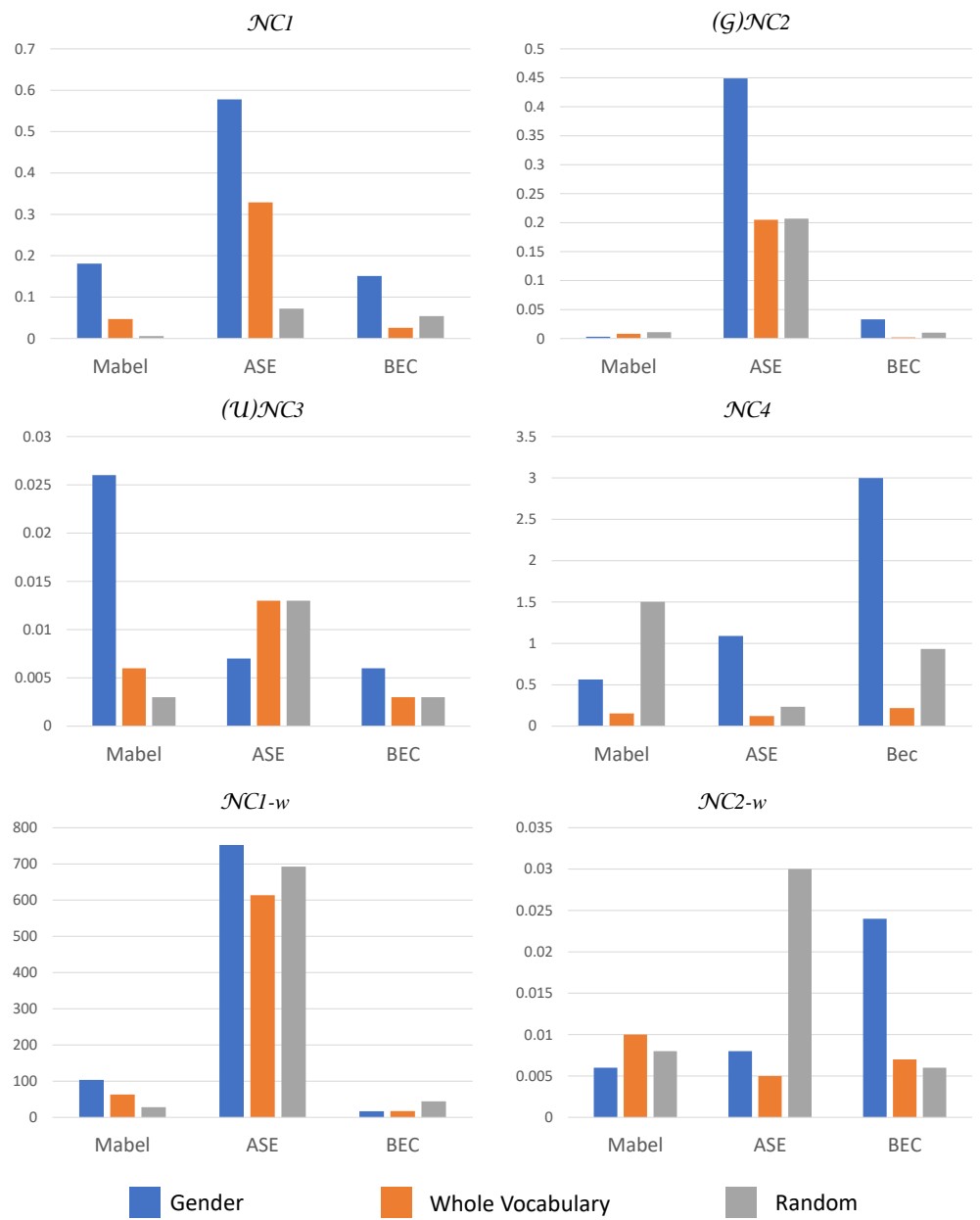

Figure 3: Absolute changes of $\mathcal{NC}$ metrics after debiasing Mabel, ASE, BEC. Values used in these plots are based on Table 1, Table 2, and Table 3.

