# OpenReview forum: "Collapsed Language Models Promote Fairness"
_ICLR.cc/2025/Conference — ICLR 2025 Poster_

### Official Review · Reviewer_6ic8 · 2024-11-03

**Soundness:** 4
**Presentation:** 3
**Contribution:** 3
**Rating:** 8
**Confidence:** 3

**Summary:**

This paper investigates the relationship between the neural collapse phenomena and fairness in language models. The paper demonstrates that debiased language models naturally exhibit more "collapsed" representations for fairness-sensitive words, and leverage this insight to develop a neural collapse-based regularization method. The proposed approach enhances fairness across different debiasing methods while maintaining model performance on general language tasks.

**Strengths:**

1.the paper draws the connection between neural collapse theory and fairness in language models in a principal way. It provides novel analytical tools for understanding debiasing mechanisms and offers new insights into how representation geometry relates to model fairness.

2.  Comprehensive empirical experiments across multiple models and datasets, demonstrating the practical utility of the proposed approach.

3. The proposed regularization method seems to be simple yet effective. It can be easily integrated with existing debiasing approaches without significant computational overhead, making it highly practical for real-world applications.

3. The empirical evaluation is thorough, covering both intrinsic and extrinsic fairness metrics. The results clearly demonstrate improvements across various benchmarks while preserving model performance on downstream tasks.

5. The authors also provide helpful visualizations and analyses that illuminate how their method affects model representations.

**Weaknesses:**

1. The primary weakness of the paper lies in its theoretical foundation. While the empirical results are promising, the authors don't provide sufficient theoretical justification for why NC3 and calibrated metrics work better than others. The paper would benefit from a formal analysis or proof connecting neural collapse to fairness improvements.

2. The scope of the evaluation is somewhat limited, focusing primarily on **binary** gender bias. While this allows for deep analysis in one area, it leaves questions about the method's generalizability to other (possibly non-binary) demographic groups.

**Questions:**

1. Can this approach generalize to non-binary protected attributes? If so, how?

2. How does the method work for decoder-only models, like gpt and llama?

---

> ### Author Response · Authors · 2024-11-21
> **Thanks for your review!**
>
> We truly thank reviewer 6ic8's time and effort in reviewing our paper!
>
> > **Q1:** More comprehensive theoretical explanation.
>
> Thanks for this suggestion! We will further consider using the layer-peeled model [1] to analyze neural collapse in language models in the context of fairness.
>
>
> > **Q2:** Regarding non-binary protected attributes.
>
> Our approach is directly applicable to non-binary protected attributes. The challenge is only from the data collection and evaluation side.
>
> Most related works (that we compare with) only study lists of binary genders of only 'female' and 'male' elements. If better word lists beyond binary genders or other non-binary attributes, and corresponding evaluation metrics become available in the future, we would be happy to conduct related experiments.
>
> > **Q3:** Regarding for decoder-only models.
>
> Thanks for this great suggestion!
>
> It is possible to extend our work to decoder-only models.
>
> However, as mentioned in paper [2], which provided a thorough summary of fairness in language models. For decoder-only models, prior methods have primarily focused on Llama or GPT. Moreover, most previous fairness and debiasing approaches have relied on prompt tuning [3,4], rather than model fine-tuning. Improvements in fairness achieved through these prompt-based methods cannot be explained by neural collapse, as they do not change the model's weights.
>
> Due to the limited timeframe, we are unable to test this extension during the rebuttal period, but we will try to add it to the final version if the paper is accepted.
> If more debiasing approaches for decoder-only models are proposed in the future, we will also be happy to conduct further research.
>
> [1] Zhu Z, Ding T, Zhou J, Li X, You C, Sulam J, Qu Q. A geometric analysis of neural collapse with unconstrained features. Advances in Neural Information Processing Systems. 2021 Dec 6;34:29820-34.
>
> [2] Yingji Li, Mengnan Du, Rui Song, Xin Wang, and Ying Wang. A survey on fairness in large language models. arXiv preprint arXiv:2308.10149, 2023a.
>
> [3] A. Tamkin, A. Askell, L. Lovitt, E. Durmus, N. Joseph, S. Kravec, K. Nguyen, J. Kaplan, D. Ganguli, Evaluating and mitigating discrimination in language model decisions, arXiv preprint arXiv:2312.03689.
>
> [4] J. Mattern, Z. Jin, M. Sachan, R. Mihalcea, B. Scholkopf, Understanding stereotypes in language models: Towards robust measurement and ¨ zero-shot debiasing, CoRR abs/2212.10678. arXiv:2212.10678, doi:10.48550/ARXIV.2212.10678.

---

> > ### Comment · Area_Chair_HGRT · 2024-11-25
> >
> > Dear Reviewer,
> >
> > Thank you for your valuable contributions to the review process for the paper!
> > The authors have submitted their rebuttal, and I would greatly appreciate it if you could take a look and provide your response.

---

### Official Review · Reviewer_fRsu · 2024-11-05

**Soundness:** 2
**Presentation:** 2
**Contribution:** 2
**Rating:** 5
**Confidence:** 4

**Summary:**

This paper proposes a method to enhance model fairness based on the phenomenon of Neural Collapse (NC). The authors first analyze NC in the context of fairness-related words and find that debiased language models exhibit more aligned token representations with word embeddings. Inspired by this observation, they introduce a regularization method based on NC to reduce bias in gender-sensitive words during fine-tuning. The experiments demonstrate that this method can improve fairness across various debiasing algorithms while maintaining the language model's performance on standard natural language understanding tasks.

**Strengths:**

The paper is logically structured and easy to follow, with a clear progression from problem statement to method proposal and experimental validation.

**Weaknesses:**

1. In Section 4, the details of the datasets and metrics can be more concisely presented, with some details moved to the appendix. Conversely, the discussion of results is somewhat inadequate. It would be beneficial to highlight the advantages of the proposed method in comparison to the baselines.
2. Additional analysis should be included in section 4.2. From the results alone, it is observed that many outcomes with the addition of (U)NC3 have significantly decreased.
3. It is unclear whether the proposed method is also effective for LLMs.
4. It is uncertain whether the method remains effective for aspects of fairness beyond gender

**Questions:**

Clarifications Needed:
1. The σ symbol on line 161.
2. On line 235, the authors hypothesize that "noises in language data and the complexity of different fine-tuning methods make measurements of NC1/2/4 unstable and inconsistent." Could the authors elaborate on why they consider these factors to be influential?
3. Could the authors justify the rationale behind replacing class means with classifier weights in NC1/2?
4. Besides gender-related words, have the authors explored other words that might cause the model to collapse more? This experiment would complement Section 3.3.
5. I am curious about the effectiveness of the proposed method in fairness scenarios beyond gender. Firstly, what if it is challenging to obtain relevant word lists? Secondly, will the model’s general performance being affected by other word lists? I think the authors could counduct experiments by adding random words/high frequency words to the word list.
6. Can the authors provide further insights into the results of the visualization in Section 5? Why do they consider this to be a better feature distribution?

---

> ### Author Response · Authors · 2024-11-21
> **Thanks for your review!  -（Part1)**
>
> We truly thank reviewer fRsu's time and effort in reviewing our paper!
>
> > **Q1:** Further discussion on the results.
>
> Thanks for the suggestion! We have included more results and discussions on BEC-Pro (Table 5) in Appendix F. We will add more discussions on our results and move some discussion from appendix to the main text in our camera ready.
>
> > **Q2:**  Clarify of Tables.
>
> Thanks for this suggestion! We have updated our Table 8 with underlines for better performance. We can see that on natural language understanding (NLU) tasks, the baseline and our method are comparable, validating that our method preserved language models’ performance on NLU tasks.
>
> > **Q3:** Regarding for Large Language Models (LLMs).
>
> Thanks for this great suggestion!
>
> It is possible to extend our work to LLMs.
>
> However, the challenge is that most previous fairness/debiasing methods for LLMs have been dominated by prompt tuning [4,5], instead of model fine-tuning, as pointed out by the survey paper [3] on fairness in language models. Improvements over fairness by these prompt tuning methods are not related to neural collapse, as model weights are unchanged.
>
> Due to the limited timeframe, we are unable to test this extension during the rebuttal period, but we will try to add it to the final version if the paper is accepted.
> If more debiasing approaches for LLMs are proposed in the future, we will also be happy to conduct further research.
>
> > **Q4:** Why we primarily focus on gender words.
>
> We mainly focus on fairness about gender because we follow previous works. Works in [6,7]  primarily focused on gender or extensively explored the associations between other words and gender. Additionally, a relatively comprehensive framework for evaluating bias [8] in gender-related words was established, allowing us to clearly observe whether NC is effective in debiasing.
>
> > **Q5:** Clarify of Notations.
>
> Thanks for pointing out! We have updated our draft with the definition of $\sigma$ in Sec. 3.1.
>
> > **Q6:** The rationale behind using classifier weights and the exclusivity of NC3's effectiveness.
>
> As explained in our Sec. 3.2.1 (Line 209-215), the neural collapse behavior manifests under certain conditions, including zero training loss and clean labels with balanced classes. However, these conditions deviate more in language model fine-tuning than vision models.
>
> Such a difference between the language model and the vision model has a direct consequence: The model predictions align more with classifier weight instead of class means in language models. This phenomenon highlights a key difference between:
>
> - **Language models**: class means are noisy and deviate from classifiers, as shown in Fig.1 (right) in [1] - the “Classifier **Agreement**” is less than 40%
>
> vs.
>
> - **Vision models**: class means are reliable and less noisy, as shown in Fig.7 of [2] - **mismatches** between class means (${\mu}_c$) and classifiers (${w}_c$) are **less than 10%**
>
> This is why we use classifier weight instead of class means in the denominator in NC1(w) for language models.
>
> Moreover, we further measure the token-prediction accuracy using either class means ($\underset{c \in \mathbb{V}}{argmin} \|{h} - {\mu}_c \|_2$) or classifiers ($\underset{c \in \mathbb{V}}{argmax} \langle{w}_c, {h} \rangle + b_c$) for language models. As shown in the table below, using class means leads to much worse predictions. These two observations motivate us to replace class means with classifiers in NC1.
>
> |        Dataset:Tinystory       | Classifier | Class Means |
> |:-------------------------|:------|:------|
> | Mabel | 7.446 | 5.064 |
> | BEC |  7.924 | 6.749 |
> |

---

> ### Author Response · Authors · 2024-11-21
> **Thanks for your review! - (Part2)**
>
> > **Q7:** How the vocabulary is obtained and the impact of random words.
>
> First, as we considered using gender words provided by the ASE work, we are not able to tell the precise efforts of collecting the list of sensitive words like those for gender. We will reach out to authors of ASE for details instructions on collecting sensitive words and estimate if this process is challenging or not.
>
> Second, following your suggestion, we also randomly added 100 high-frequency words and 100 low-frequency words to the gender word list. The total amounts of random words we added (200) is approximately 50% of the number of gender words in the original list (425, see Appendix A). The results with these additional words are shown in the table below. We can observe that adding unrelated words has a negative impact on the model's bias.
>
> | Model                     | Female   | Male     | Diff (smaller the better)   |
> |---------------------------|----------|----------|---------|
> | ASE                       | -0.7534  | -0.5125  | **0.2409** |
> | ASE + random words        | -0.7587  | -0.3851  | 0.3736 (+0.1327) |
> | ASE + NC3                 | -1.0093  | -0.9613  | **0.0480** |
> | ASE + NC3 + random words  | -1.0813  | -0.8255  | 0.2558 (+0.2078)  |
> |
>
> > **Q8:** Explanation of the visualizations in Section 5.
>
> From Figure 2 in the main text, we can observe that when we compare the gender word clusters with and without adding $L_{nc3}$ to the training, the word embeddings within each class become more collapsing together. This aligns with the definition of NC3 and provides evidence that our training is effective.
>
> Moreover, Reviewer also acknowledged our visualization: “provide helpful visualizations and analyses that illuminate how their method affects model representations.”
>
>
> [1] Wu R, Papyan V. Linguistic Collapse: neural collapse in (Large) Language Models. arXiv preprint arXiv:2405.17767. 2024 May 28.
>
> [2] Papyan V, Han XY, Donoho DL. Prevalence of neural collapse during the terminal phase of deep learning training. Proceedings of the National Academy of Sciences. 2020 Oct 6;117(40):24652-63.
>
> [3] Yingji Li, Mengnan Du, Rui Song, Xin Wang, and Ying Wang. A survey on fairness in large language models. arXiv preprint arXiv:2308.10149, 2023a.
>
> [4] A. Tamkin, A. Askell, L. Lovitt, E. Durmus, N. Joseph, S. Kravec, K. Nguyen, J. Kaplan, D. Ganguli, Evaluating and mitigating discrimination in language model decisions, arXiv preprint arXiv:2312.03689.
>
> [5] J. Mattern, Z. Jin, M. Sachan, R. Mihalcea, B. Scholkopf, Understanding stereotypes in language models: Towards robust measurement and ¨ zero-shot debiasing, CoRR abs/2212.10678. arXiv:2212.10678, doi:10.48550/ARXIV.2212.10678.
>
> [6] Jacqueline He, Mengzhou Xia, Christiane Fellbaum, and Danqi Chen. Mabel: Attenuating gender bias using textual entailment data. arXiv preprint arXiv:2210.14975, 2022
>
> [7] Bartl, Marion, Malvina Nissim, and Albert Gatt. "Unmasking contextual stereotypes: Measuring and mitigating BERT's gender bias." arXiv preprint arXiv:2010.14534 (2020).
>
> [8] M. De-Arteaga, A. Romanov, H. M. Wallach, J. T. Chayes, C. Borgs, A. Chouldechova, S. C. Geyik, K. Kenthapadi, A. T. Kalai, Bias in bios: A case study of semantic representation bias in a high-stakes setting, in: Proceedings of the Conference on Fairness, Accountability, and Transparency, FAT, 2019, pp. 120–128.

---

> > ### Author Response · Authors · 2024-11-25
> > **Look forward to more discussions**
> >
> > Dear Reviewer fRsu.
> >
> > As the author-reviewer discussion period is nearing its end, and since other reviewes have actively engaged in discussions, we would greatly appreciate it if you could review our responsesto your comments at your earliest convenience.
> >
> > This will allow us to address any further questions or concerns you may have beforethe discussion period concludes. If our responses satisfactorily address your concerns, we kindly ask you to considerevising your rating of our work.
> >
> > Thank you very much for your time and effort!
> >
> > Sincerely,
> >
> > The Authors of Submission #281

---

> > > ### Comment · Area_Chair_HGRT · 2024-11-25
> > >
> > > Dear Reviewer,
> > >
> > > Thank you for your valuable contributions to the review process for the paper!
> > > The authors have submitted their rebuttal, and I would greatly appreciate it if you could take a look and provide your response.

---

> > > ### Author Response · Authors · 2024-11-29
> > > **Look forward to more discussions**
> > >
> > > Dear Reviewer fRsu,
> > >
> > > As the author-reviewer discussion period is approaching its deadline, we kindly ask if you could review our responses to your comments at your earliest convenience. We value your feedback and want to ensure we address any remaining questions or concerns you may have before the discussion period concludes.
> > >
> > > If there are any further points that require clarification or additional information, please do not hesitate to reach out. We are happy to provide further details and would greatly appreciate your continued engagement. If our responses satisfactorily address your concerns, we would be grateful if you could consider revising your rating of our work.
> > >
> > > Thank you once again for your time and thoughtful review!
> > >
> > > Sincerely,
> > >
> > > The Authors of Submission #281

---

### Official Review · Reviewer_adpT · 2024-11-06

**Soundness:** 3
**Presentation:** 3
**Contribution:** 2
**Rating:** 8
**Confidence:** 3

**Summary:**

### Summary

- The paper studies the relationship between neural collapse and fairness and tries to answer two research questions
	- "Do debiased LMs exhibit greater neural collapse"
	- "Can this inductive bias be leveraged to improve fairness"
- Background on neural collapse:
	- This is a phenomenon that happens in the last layer representations of classifiers.
	- Since the last layer of LM predicts the next word, it can be viewed as a classifier
	- The phenomenon of neural collapse causes the representations of certain gender sensitive words to collapse closer to their class means in debiased LMs
- The paper presents 4 ways to quantify neural collapse borrowing it from prior work. These metrics measure separability between classes.
	- NC1: intra-class variance over inter class variance
	- NC2: Measures separability from a geometric perspective
	- NC3: Uses dot products
	- NC4: Uses a linear projection

In section 3.2.1 the paper provides a worry about diverging values for various NC metrics but convincingly provides a resolution. This is achieved by calibrating the NC metrics by replacing class means with classifier weights in the expressions.
- The experimental results justify the claim of the presence of neural collapse for gender words.
- The second part of the paper deals with proposing a bias mitigation strategy by explicitly enforcing Neural collapse. To this end, they use the NC3 formulation as an auxiliary objective during fine-tuning.
- Both intrinsic and extrinsic metrics show improvements with this modified objective.

Please consider citing the following additional work in the field of LM fairness and evaluation of bias in LMs
- https://arxiv.org/pdf/2208.01448#page=14.61
- https://aclanthology.org/2022.findings-acl.55.pdf
- https://aclanthology.org/2022.acl-long.401.pdf

nits
- 3.2.2 CALIBRATIONS -> CALIBRATION
- Section 4.1.3
	- TPR-1 (Type 1: pro-stereotypical minus anti-stereotypical) and TPR-2 (Type 2: pro-stereotypical minus anti-stereotypical) -> TPR-1 (Type 1: pro-stereotypical minus anti-stereotypical) and TPR-2 (Type 2: pro-stereotypical minus anti-stereotypical) reads a bit odd. I think Type 1 are ambiguous sentences and Type 2 are non-ambiguous sentences. Explicitly call this out with a simple example.

**Strengths:**

### Strengths

- The paper studies a missing link between neural collapse and fairness and proposes a bias mitigation strategy based on neural collapse objective that is agnostic to pre-training or fine-tuning methods. This also avoids manual data balancing or filtering.
- Both intrinsic and extrinsic fairness metrics see improvements with this approach.

**Weaknesses:**

### Weakness

- Table 9 does not have the best results bold which makes it reading a little harder.
- While the paper is well-written, simple to follow and proposes a simple technique which works, all the experiments conducted are on BERT based MLM based models. I understand this is intentional to ensure fair comparison with prior work but this also limits understanding the degradation on non MLM tasks. For eg. in Table 9, the results reported are on the GLUE benchmark which are all multiple choice questions. I am curious if this technique can really prove to be a generic technique even for generation tasks.

**Questions:**

No questions, see weakness section above.

---

> ### Author Response · Authors · 2024-11-21
> **Thanks for your review!**
>
> We truly thank reviewer adpT's time and effort in reviewing our paper!
>
> > **Q1:** Clarify of Tables.
>
> Thanks for the suggestion! Table 9 had bold for the best results in our previous submission. We have revised our Table 8 accordingly with underlines
>
> > **Q1:** Regarding for generation tasks.
>
> Thanks for this great suggestion!
>
> It is possible to extend our work to generation tasks.
>
> However, the challenge is that most previous fairness/debiasing methods for generation tasks have been dominated by prompt tuning [1,2], instead of model fine-tuning, as pointed out by the survey paper [3] on fairness in language models. Improvements over fairness by these prompt tuning methods are not related to neural collapse, as model weights are unchanged.
>
> Due to the limited timeframe, we are unable to test this extension during the rebuttal period, but we will try to add it to the final version if the paper is accepted.
> If more debiasing approaches for generation tasks are proposed in the future, we will also be happy to conduct further research.
>
> [1] A. Tamkin, A. Askell, L. Lovitt, E. Durmus, N. Joseph, S. Kravec, K. Nguyen, J. Kaplan, D. Ganguli, Evaluating and mitigating discrimination in language model decisions, arXiv preprint arXiv:2312.03689.
>
> [2] J. Mattern, Z. Jin, M. Sachan, R. Mihalcea, B. Scholkopf, Understanding stereotypes in language models: Towards robust measurement and ¨ zero-shot debiasing, CoRR abs/2212.10678. arXiv:2212.10678, doi:10.48550/ARXIV.2212.10678.
>
> [3] Yingji Li, Mengnan Du, Rui Song, Xin Wang, and Ying Wang. A survey on fairness in large language models. arXiv preprint arXiv:2308.10149, 2023a.

---

> > ### Comment · Reviewer_adpT · 2024-11-24
> > **Response**
> >
> > Thank you to the authors for their response. I did not have many concerns, and I also reviewed other reviewer comments, keeping my score at 8.

---

### Official Review · Reviewer_ypXD · 2024-11-07

**Soundness:** 2
**Presentation:** 3
**Contribution:** 3
**Rating:** 6
**Confidence:** 3

**Summary:**

This work explores fairness in language models from the perspective of Neural Collapse (NC), a phenomenon observed during the final stages of training neural networks on classification tasks (thus can also be applied to language modeling), where the variability and alignment of the penultimate layer and the final classifier reach a structured state. This work further applies NC to debiased LMs and demonstrates that  $\mathcal{NC}3$, which measures the alignment between token representations (contextual class means) and word embeddings (classifiers), consistently improved with debiasing. The author argues that other NC perspectives are less evident in debiased LMs due to common training configurations that violate NC’s typical conditions—such as vocabulary sizes exceeding embedding dimensions and imbalanced token distributions. To address this, they propose calibrated versions of  $\mathcal{NC}1$ and  $\mathcal{NC}2$ by replacing class means with classifier weights. Debiased LM shows significant collapse on these two metrics, and fairness-sensitive words demonstrate more collapse.

Based on the empirical analysis, the authors suggest using  $\mathcal{NC}3$ as a regularization term during fine-tuning to enhance fairness in LMs. Experiments on four fairness datasets demonstrate consistent gains on fairness metrics over three prior debiasing approaches and standard masked language modeling and also doesn't hurt the model's NLU ability.

**Strengths:**

1. Analyzing the debiased language models from the perspective of neural collapse is novel. This offers valuable insights into the structure of the model's embedding space across different debiasing approaches.
2. The proposed method is simple but effective and can be easily integrated with other approaches. It's interesting to see that by only adding a NC-based regularization term in the loss function, all the compared debiasing methods can be enhanced.
3. The experiments are comprehensive and well-motivated, with the proposed method grounded in observations of $\mathcal{NC}$ perspectives on both biased and debiased LMs, effectively establishing a strong rationale for the proposed method, making it clear and convincing.

**Weaknesses:**

1. **Clarify of Notations**: Some notations are not clearly defined, which could impact readability.
    - The class embedding variances mentioned on line 161 and line 240 could benefit from a more explicit definition, similar to how it is presented in Eq. 3 of the Linguistic Collapse[1] paper.
    - In Eq. 1, to align with the token representation defined at line 137, the notation should ideally be ${h}(E(x_{1:t}))$ rather than ${h}(x_{1:t})$ for clarity and consistency.
2. **Evaluation Metrics**: I have concerns regarding the evaluation metrics used in section 3.3 and section 4.2.1. Please see question 2 and 3 in the following section.
3. **Definition of Unfairness**: Some definition of unfairness were not explicitly defined, which could make the paper less accessible to readers unfamiliar with the fairness literature. For example, in the *BEC-Pro* subsection of section 4.1.2, the average association score is used to measure model unfairness. While this is defined in [2] as $\log \frac{P_T}{P_\text{prior}}$, it's not explicitly explained here, which may cause confusion.

[1] Wu, Robert, and Vardan Papyan. "Linguistic Collapse: Neural Collapse in (Large) Language Models." arXiv preprint arXiv:2405.17767 (2024).
[2] Bartl, Marion, Malvina Nissim, and Albert Gatt. "Unmasking contextual stereotypes: Measuring and mitigating BERT's gender bias." arXiv preprint arXiv:2010.14534 (2020).

**Questions:**

1. In section 3.2.2, the authors present a calibrated version of $\mathcal{NC}1$ and  $\mathcal{NC}2$. While I can understand that the main reason to replace class means with classifier weights is because the authors find that $\mathcal{NC}3$ (solely derived from classifier weights) is significant on the debiased models, could this modification compromise the original physical meaning of these metrics Specifically,
    - $\mathcal{NC}1$, as proposed in [1], measures the variability via an inverse signal-to-noise ratio. However, in the calibrated $\mathcal{N}\mathcal{C}_{1}^{(w)}$, the numerator is the variance of the _class means_, while the denominator measures the distance of the _classifier weight_, which seems to lack physical coherence.
    - Why is $w_c$ not normalized in $\mathcal{(G)NC_2^{(w)}}$ while $\mu_c$ is normalized in $\mathcal{(G)NC_2}$ to measure uniformity?
    - I hope the authors can elaborate more on the reason they choose these calibrated metrics.
2. In section 3.3, the authors argue that _"debiased BERT models exhibit more different token representations and word embeddings only on gender-related vocabulary"_ based on __numerical gaps__ in $\mathcal{NC}$ metrics. However, the baseline level of the base BERT model differ across gender-related words, full vocabulary, and a randomly selected subset. By looking at the __ratios__ of the gap w.r.t. base BERT model's performance under $\mathcal{NC_1^{(w)}}$, I cannot tell there are significant differences among Table 1-3. Similarly, with $\mathcal{(U)NC_3}$, for the gap of ASE and BEC, it's still hard to tell the difference among these three tables.
3. In section 4.1.2, could the authors clarify why the difference in association scores between female and male groups is measured? If the association score is defined as $\log \frac{P_T}{P_\text{prior}}$, I feel it's more reasonable to measure the absolute value of the association score since if the absolute value is close to 0, it should suggest that <person word> predictions are less likely influenced by <profession>, which indicates a less biased LM.
4. All experiments in this paper are conducted on masked language models (e.g, BERT and RoBERTa). Could the proposed method also generalize to casual language models, as explored in the Linguistic Collapse[1] paper?

[1] Wu, Robert, and Vardan Papyan. "Linguistic Collapse: Neural Collapse in (Large) Language Models." arXiv preprint arXiv:2405.17767 (2024).

---

> ### Author Response · Authors · 2024-11-21
> **Thanks for your review! - (Part1)**
>
> We truly thank reviewer ypXD's time and effort in reviewing our paper!
>
> > **Q1:** Clarify of Notations.
>
> Thanks for these comments! We have revised these notations in blue in our updated draft (Eq. 3 on page 3).
>
> > **Q2:** The reason we use classifiers instead of class means.
>
> As explained in our Sec. 3.2.1 (Line 209-215), the neural collapse behavior manifests under certain conditions, including zero training loss and clean labels with balanced classes. However, these conditions deviate more in language model fine-tuning than vision models.
>
> Such a difference between the language model and the vision model has a direct consequence: The model predictions align more with classifier weight instead of class means in language models. This phenomenon highlights a key difference between:
>
> - **Language models**: class means are noisy and deviate from classifiers, as shown in Fig.1 (right) in [1] - the “Classifier **Agreement**” is less than 40%
>
> vs.
>
> - **Vision models**: class means are reliable and less noisy, as shown in Fig.7 of [2] - **mismatches** between class means (${\mu}_c$) and classifiers (${w}_c$) are **less than 10%**
>
> This is why we use classifier weight instead of class means in the denominator in NC1(w) for language models.
>
> Moreover, we further measure the token-prediction accuracy using either class means ($\underset{c \in \mathbb{V}}{argmin} \|{h} - {\mu}_c \|_2$) or classifiers ($\underset{c \in \mathbb{V}}{argmax} \langle{w}_c, {h} \rangle + b_c$) for language models. As shown in the table below, using class means leads to much worse predictions. These two observations motivate us to replace class means with classifiers in NC1.
>
> |        Dataset:Tinystory       | Classifier | Class Means |
> |:-------------------------|:------|:------|
> | Mabel | 7.446 | 5.064 |
> | BEC |  7.924 | 6.749 |
> |
>
>
> > **Q3:** Comparison of using and not using normalization on (G)NC2(w).
>
> For (G)NC2(w), we studied both with and without normalization. As shown in table below, we found that the impact of normalization was minimal and will not change our conclusion.
>
> | Model | (G)Nc2(w) | (G)Nc2(w-norm) |
> |-------|--------|------------|
> | BERT  | 0.337  | 0.170      |
> | Mabel | 0.331  | 0.167      |
> |
> | BERT  | 0.364  | 0.139      |
> | ASE   | 0.372  | 0.135      |
> |
> | BERT  | 0.359  | 0.143      |
> | BEC   | 0.355  | 0.142      |
> |
>
> > **Q4:** Visualizing the gaps of NC across different word lists.
>
> Thanks for your comment!
>
> Our main conclusion is that: debiased language models collapse more on fairness-sensitive words (instead of the whole vocabulary and random words). The significant differences between Table 1 vs. Table 2 or 3 support the conclusion. Indeed, based on your suggestion, we can illustrate this conclusion with a better presentation. In the table below, we show absolute differences (on NC1) between the biased model and the debiased model. We can clearly see more significant gaps on gender words (than whole vocabulary and random words).
>
> | NC1_debiased - NC1_biased | Mabel  | ASE    | BEC   |
> |---------------------------|--------|--------|--------|
> | Gender                   | **0.181** | **0.578** | **0.151** |
> | Whole Vocabulary         | 0.047  | 0.329  | 0.026  |
> | Random Words             | 0.006  | 0.072  | 0.054  |
> |
>
> In our revised paper, we also show a bar plot of this Table in Appendix G for better illustration.

---

> ### Author Response · Authors · 2024-11-21
> **Thanks for your review! - (Part2)**
>
> > **Q5:** Why we choose to use the difference in association scores between female and male on the BEC-pro.
>
> We greatly appreciate your valuable feedback, which is a meaningful complement to our work. We think that both your comment and our explanation are correct, corresponding to two different but reasonable fairness definitions in this context.
>
> 1. Your interpretation of the association score is correct. It measures the correlation between a `<person word>` and a `<profession>` (smaller the less correlated). However, the association score itself is not an indicator of gender bias. For example, the associations of male and female could be both small, but any of their difference indicates a bias over genders (i.e. the profession has different inclinations to male and female).
> 2. When we measure the difference in the association scores between `<female>` and `<male>` with respect to a `<profession>`, this represents gender bias when associating with professions, thus serving as a more appropriate intrinsic measure, one that is specifically related to gender.
>
> To facilitate your understanding, we have included an additional experiment below, where we also optimize NC3 in BEC using different sets of words. As shown in the table below, using gender or profession words can both reduce the association scores, but only regularizing NC3 on gender words can make the association with female and male less distinguishable (reduced diff), whereas regularizing on profession actually enlarges the gender bias.
>
> |       Model      | Female   | Male     | Diff      |
> |------------------|----------|----------|-----------|
> | BEC              | 0.0841   | 0.1349   | 0.0508    |
> | BEC + NC3 on gender ($\alpha=10$) | 0.0632 (-0.0209) | 0.1103 (-0.0246) | **0.0471 (-0.0037)** |
> | BEC + NC3 on profession ($\alpha=10$) | 0.0429 (-0.0412) | 0.1200 (-0.0149) | **0.0771 (+0.0263)** |
> |
>
>
> > **Q6:** Regarding causal language models.
>
> Thanks for this great suggestion!
>
> It is possible to extend our work to causal language models.
>
> However, the challenge is that most previous fairness/debiasing methods for causal language models have been dominated by prompt tuning [4,5], instead of model fine-tuning, as pointed out by the survey paper [3] on fairness in language models. Improvements over fairness by these prompt tuning methods are not related to neural collapse, as model weights are unchanged.
>
> Due to the limited timeframe, we are unable to test this extension during the rebuttal period, but we will try to add it to the final version if the paper is accepted.
> If more debiasing approaches for causal language models are proposed in the future, we will also be happy to conduct further research.
>
>
> [1] Wu R, Papyan V. Linguistic Collapse: neural collapse in (Large) Language Models. arXiv preprint arXiv:2405.17767. 2024 May 28.
>
> [2] Papyan V, Han XY, Donoho DL. Prevalence of neural collapse during the terminal phase of deep learning training. Proceedings of the National Academy of Sciences. 2020 Oct 6;117(40):24652-63.
>
> [3] Yingji Li, Mengnan Du, Rui Song, Xin Wang, and Ying Wang. A survey on fairness in large language models. arXiv preprint arXiv:2308.10149, 2023a.
>
> [4] A. Tamkin, A. Askell, L. Lovitt, E. Durmus, N. Joseph, S. Kravec, K. Nguyen, J. Kaplan, D. Ganguli, Evaluating and mitigating discrimination in language model decisions, arXiv preprint arXiv:2312.03689.
>
> [5] J. Mattern, Z. Jin, M. Sachan, R. Mihalcea, B. Scholkopf, Understanding stereotypes in language models: Towards robust measurement and ¨ zero-shot debiasing, CoRR abs/2212.10678. arXiv:2212.10678, doi:10.48550/ARXIV.2212.10678.

---

> > ### Author Response · Authors · 2024-11-25
> > **Look forward to more discussions**
> >
> > Dear Reviewer ypXD.
> >
> > As the author-reviewer discussion period is nearing its end, and since other reviewes have actively engaged in discussions, we would greatly appreciate it if you could review our responsesto your comments at your earliest convenience.
> >
> > This will allow us to address any further questions or concerns you may have beforethe discussion period concludes. If our responses satisfactorily address your concerns, we kindly ask you to considerevising your rating of our work.
> >
> > Thank you very much for your time and effort!
> >
> > Sincerely,
> >
> > The Authors of Submission #281

---

> > > ### Comment · Area_Chair_HGRT · 2024-11-25
> > >
> > > Dear Reviewer,
> > >
> > > Thank you for your valuable contributions to the review process for the paper!
> > > The authors have submitted their rebuttal, and I would greatly appreciate it if you could take a look and provide your response.

---

> > ### Comment · Reviewer_ypXD · 2024-11-26
> > **Thank you for your detailed response**
> >
> > Thank you for providing thoughtful and detailed responses to my questions! Your clarifications and the revision of the paper help me better understand the contribution of this work. I have increased my score from 5 to 6.
> >
> > One more question regarding Q2 in my original review, which was not fully addressed in Q4 of your rebuttal. Perhaps I am misunderstanding, but why do you consider the **numerical absolute gap** between the biased base model and the debiased model (rather than the **ratio** of the difference), given that the biased base model exhibits significantly different performance across Tables 1/2/3 to support your statement regarding `line 256`? Specifically, as the NC metrics in Table 1 are generally larger than those in Tables 2/3, could the larger gap in Table 1 simply result from the biased base model learning less separable representations of gender words? Starting from this, I suggests the authors to consider including another ablation study to the final version by applying the proposed $L_{NC_3}$ to other vocabulary subsets other than the gender words (e.g., the neutral words) and comparing the results with the current design in Eq. 3 to better support the decision of applying $L_{NC_3}$ specifically to gender words.

---

> > > ### Author Response · Authors · 2024-11-27
> > > **Thanks for your review!**
> > >
> > > Thank you very much for your valuable feedback.
> > >
> > > > **Q1:**  Why use the absolute value to measure this gap?
> > >
> > > Absolute differences are typically used when measurements are in the same unit with consistent scales.
> > > Our Tables 1, 2, and 3 calculate the same sets of metrics over word lists of the same length, differing only in the specific words used in their lists.
> > > This suggests that the measurement scale is consistent across these tables.
> > > Furthermore, when training with $L_{NC3}$, we use the absolute values of NC3, which guide the adjustment of the hyperparameter $\alpha$.
> > > Therefore, we believe that using absolute values to measure the gap is more appropriate.
> > >
> > > > **Q2:**  Adding other vocabulary subsets.
> > >
> > > We sincerely appreciate your comments. Similar concerns were raised by other reviewers earlier, and we would like to provide further clarification here.
> > >
> > > We randomly added 100 high-frequency words and 100 low-frequency words to the gender word list, which were obtained from ASE. The total amounts of random words we added (200) is approximately 50% of the number of gender words in the original list (425, see Appendix A). The results with these additional words are shown in the table below. We can observe that adding unrelated words has a negative impact on the model's bias.
> > >
> > > | Model                     | Female   | Male     | Diff (smaller the better)   |
> > > |---------------------------|----------|----------|---------|
> > > | ASE                       | -0.7534  | -0.5125  | **0.2409** |
> > > | ASE + random words        | -0.7587  | -0.3851  | 0.3736 (+0.1327) |
> > > | ASE + NC3                 | -1.0093  | -0.9613  | **0.0480** |
> > > | ASE + NC3 + random words  | -1.0813  | -0.8255  | 0.2558 (+0.2078)  |
> > > |

---

### Author Response · Authors · 2024-11-21
**We thank comments, questions, and suggestions by all reviewers!**

We deeply appreciate the feedback and suggestions from all four reviewers. We are pleased that all four reviewers acknowledged our main contribution and recognized our: 1) novelty and insights into neural collapse and fairness; 2) simplicity and effectiveness of our proposed method; 3) comprehensive and well-executed experiments; 4) clarity and logical presentation; 5) improvement across metrics.

We address all questions and concerns in individual responses. Following the ICLR guidelines, we also updated our draft with revised parts in blue.

---

### Meta-Review · Area_Chair_HGRT · 2024-12-22

**Metareview:**

(a) **summary** This paper investigates the connection between Neural Collapse (NC) -- a learning phenomenon that happens in last-layer representations and classifiers in deep networks -- and fairness in language models (LMs). The authors hypothesize that similar effects may manifest in masked language models (MLMs), particularly for fairness-sensitive (e.g., gender-related) words, with two findings: (1) Debiased LMs show stronger Neural Collapse for sensitive words, (2) It is possible to leverage NC to improve fairness.

(b) **strengths**: The paper proposes a simple yet effective approach for improving fairness in LMs based on the observations of NC, and the evaluation is comprehensive

(c) **weakness**: (1) some reviewers request a formal analysis on why NC3 specifically underpins fairness improvements better than other NC metrics or calibrated versions (NC1/2/4); (2) lack of clarity in notations and definitions (3) narrow scope, which only focuses on gender bias and masked LM. It seems unclear whether it can be applied to other types of bias as well as decoder-only models

(d) **decision** Overall, the reviewers find the paper sound in its empirical demonstration and novel in its perspective connecting Neural Collapse to fairness. The method is straightforward to incorporate into existing debiasing setups and consistently yields fairness improvements. While the paper would benefit from more thorough theoretical grounding and clarifications in notations, given the paper’s novelty, promising empirical results, and relatively simple yet effective approach, I recommend acceptance.

**Additional Comments On Reviewer Discussion:**

ypXD: The author solved the reviewers' concerns on notations and other confusions, which leads to a score bump from 5 to 6.
 fRsu & 6ic8:  The reviewers are not responsive.

---

### Decision · Program_Chairs · 2025-01-22

Accept (Poster)